# Predicting evolutionary rate as a pretraining task improves genome language model representations

**Micaela Elisa Consens** [1]  **Kevin K. Yang** [2]  **Jimmy Hall** [2]  **Ashley Mae Conard** [2]  **Bo Wang** [1]  **Lorin Crawford** [2]
**Alan M. Moses** [1]  **Alex X. Lu** [2]

## Abstract

Genome language models (gLM) have the potential to further understanding of regulatory genomics without requiring labeled data. Most gLMs are pretrained using sequence reconstruction tasks inspired by natural language processing, but recent studies have shown that these gLMs often fail to capture biological signal. To overcome this, we introduce pretraining tasks that predict the rate of evolution. These tasks are designed so that they can be composed with sequence reconstruction, enabling a controlled comparison of predicting sequence only, evolutionary rate only, or both. To address gaps in existing evaluations, we developed a suite of biologically grounded benchmarks. Across these tasks, and for established variant effect prediction benchmarks, models pretrained on both sequence and evolutionary rate outperform those trained on sequence alone, and training on evolutionary rate can make the even the relatively small models in our work competitive with much larger existing gLMs for some tasks on the human genome. These results establish evolution as a key training target for genome-scale models.

## 1. Introduction

Genome language models (gLMs) are models pretrained with a self-supervised objective on genomic sequences. Amongst other applications, gLMs promise to learn how and when genes are activated, repressed, or modulated by their regulatory context. Learning this "grammar" would be particularly impactful for human gene regulation, enabling advances in personalized medicine, drug discovery, and synthetic biology. Thus, there has been a rapid increase in the number of proposed gLMs along with the financial scale of these efforts (Nguyen et al., 2023; Schiff et al., 2024; Benegas et al., 2024; Dalla-Torre et al., 2025; Albors et al., 2025; Brixi et al., 2025).

However, there has been little exploration of gLM pretraining tasks. Most of gLMs are pretrained using sequence reconstruction tasks inspired by natural language processing such as predicting the next token given previous tokens (i.e., next token prediction (NTP)) or predicting masked tokens from surrounding context (i.e., masked language modeling (MLM)) (Figure 1B). While these models have demonstrated strong performance after fine-tuning on tasks like predicting transcription factor binding, chromatin profile activity, and regulatory region identification (Dalla-Torre et al., 2025), recent works show that they underperform simple baselines (Tang et al., 2025) and do not reliably outperform randomly-initialized versions of their architectures (Vishniakov et al., 2024). Additionally, few works have assessed models in the zero-shot setting, which is particularly important for biological understanding applications where labels may not be known.

In contrast to gLM pretraining, many bioinformatics methods for inferring genome function exploit cross-species genome comparisons. Genome databases (Perez et al., 2025; Dyer et al., 2025) have enabled construction of whole genome alignments and the direct estimation of evolutionary rates at single-base resolution. Evolutionary rate reflects functional importance: functional sites tend to be conserved, accumulating substitutions more slowly than neutral sites. Furthermore, evolutionary rate captures fine-grained function within genomic elements (e.g., for protein coding regions, degenerative positions that code for the same amino acid show lower rates). Thus, evolutionary rate remains one of the most powerful and widely applied predictors of genome function (Zhou & Troyanskaya, 2015; Consens et al., 2025; Pollard et al., 2010; Zhou et al., 2011; Rentzsch et al., 2019; Benegas et al., 2024; Albors et al., 2025) (Figure 1A). Still, despite its effectiveness in detecting function, evolutionary rate has not yet been widely investigated for

[1]Department of Computer Science, University of Toronto, Toronto, Ontario, Canada [2]Microsoft Research, Cambridge, Massachusetts, USA. Correspondence to: Micaela Elisa Consens <micaela.consens@mail.utoronto.ca>.

*Proceedings of the 43rd International Conference on Machine Learning*, Seoul, South Korea. PMLR 306, 2026. Copyright 2026 by the author(s).

training gLMs.

In this work, we evaluate evolutionary rate prediction as a pretraining task. We introduce two novel evolutionary rate prediction pretraining tasks, to predict PhyloP scores (Figure 1A). In current evolution prediction (CEP), the model predicts the evolutionary rate at each position given the sequence up to that position (Figure 1C). In masked evolution modeling (MEM), the model is given a partially masked nucleotide sequence and learns to predict evolutionary rates at all positions (Figure 1D). Notably, these tasks are composable with NTP and MLM, respectively, and only require single sequences as inputs (Figure 1B). To compare pretraining strategies, we trained "Gamba" models, using the same architectures and training data, on sequence reconstruction, evolutionary-rate prediction, and their combination. We then developed a biologically-aligned zero-shot benchmark and show that pretraining on evolutionary rates consistently improves representation quality, highlighting its potential as a pretraining signal composable with sequence reconstruction.

## 2. Related Works

### 2.1. Training genome language models with evolution

In principle, pretraining with multi-species sequence reconstruction implicitly captures evolutionary constraints. By encountering homologous genomic regions across species, gLMs can learn conserved patterns that reflect functional regions, analogous to how protein language models trained on single sequences from across species capture functional and structural constraints (Zhang et al., 2024). Indeed, pretraining on evolutionarily diverse sequences improves the capabilities of gLMs in modeling functional relevance (Dalla-Torre et al., 2025; Consens et al., 2025; Brixi et al., 2025) as opposed to human-only pretraining. However, genomes are significantly noisier than protein sequences, and are dominated by "junk DNA" of unknown functional significance (Fagundes et al., 2022). It is unclear whether presenting models with genomes across species is as effective a strategy for implicitly learning evolution in genomes as it has been for proteins. Furthermore, genomes are much longer than proteins, making it more enticing to compress evolutionary information from related species into a simple target, and unlike in protein modeling, we are often primarily interested in the human genome. Thus, in our work, we assess the effectiveness of directly predicting evolution.

One way to directly incorporate evolutionary signal **explicitly** during pretraining is by reconstructing MSAs instead of single sequences. GPN-MSA is a masked language model over MSAs (Benegas et al., 2024) with strong performance on non-coding variant effect prediction tasks. GPN-STAR improves performance on non-coding variant effect predic-

tion further by using the phylogenetic tree inferred from the MSA to constrain attention between sequences (Ye et al., 2025). However, requiring MSAs as inputs limits downstream applications (e.g., synthetic sequences, which will not have natural homologs in other species).

Deep learning approaches for predicting evolutionary rate have been developed previously for specific tasks, such as the DeepCons CNN-based model for identifying functional regions (Li et al., 2017) and the GPN models for variant effect prediction (Benegas et al., 2024; Ye et al., 2025). Our contribution incorporates evolutionary rate prediction directly into language model pretraining objectives, enabling learned representations to be flexibly applied across diverse genomic tasks.

The closest approach is PhyloGPN, which also predicts evolutionary rate as a pretraining target (Albors et al., 2025). However, instead of predicting the rate at every position (CEP/MEM), PhyloGPN predicts the rate at a single central nucleotide per input. This is less efficient to train, and they do not evaluate if evolutionary rate-based pretraining can be composed with sequence reconstruction. In contrast, we combine CEP with NTP and MEM with MLM, allowing us to conduct ablations using the same architecture and training data to quantify the impact of training on evolutionary rate only, sequence only, or both.

### 2.2. Evaluation suites for genome language models

A key challenge for gLMs is evaluating whether pretraining captures biologically meaningful signals across diverse genomic regions. Many evaluation suites train supervised classifiers on top of frozen gLM representations which introduces variance driven by differing representation dimensionality between models and adds unnecessary computational overhead (Tang et al., 2025; Marin et al., 2023). Randomly-initialized baselines are also often absent, making it difficult to determine whether reported performance reflects learned biological signal or architectural bias (Grešová et al., 2023; Zhou et al., 2023; Dalla-Torre et al., 2025; Vishniakov et al., 2024). In addition, how genomic regions are sampled for positive and negative classes can introduce systematic biases. For example, contrasting functional regions with generic background DNA tends to reward models that exploit low-level sequence composition rather than mechanism-specific features. Most benchmarks further collapse all negatives into a single background class, obscuring which cues a model is relying on; when negatives are easy (e.g., random genomic DNA) models can achieve high accuracy by detecting broad compositional properties rather than learning regulatory or structural constraints.

To address these issues, we introduce evaluations that (i) measure zero-shot performance without training supervised classifiers, (ii) include randomly-initialized baselines

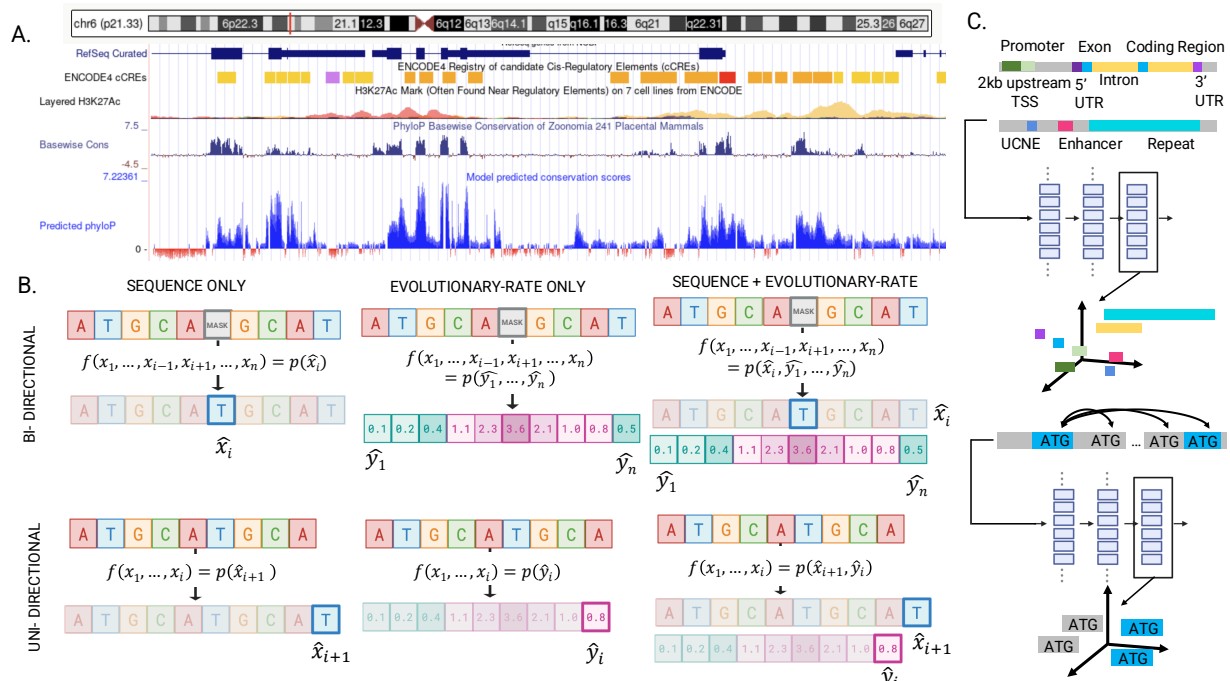

*Figure 1.* A) PhyloP scores (a measure of evolutionary rate) and our ArGamba predicted PhyloP scores B) Pretraining tasks with bi-directional sequence context (Masked language model (MLM), Masked evolution modeling (MEM), MLM combined with MEM), and tasks with autoregressive context (Next token prediction (NTP), Current evolution prediction (CEP), NTP combined with CEP). C) Our zero-shot benchmarking tasks which evaluate models' DNA representations in their high-dimensional representation space.

matched for architecture and representation size, and (iii) use multiple negative sets of systematically increasing difficulty to mitigate sampling bias and reveal what signals models actually learn.

# 3. The Gamba models

## 3.1. Current evolution prediction and masked evolution modeling

We propose two pretraining tasks that predict evolutionary rate given genomic sequence as input: current evolution prediction (CEP) and masked evolution modeling (MEM), paralleling the sequence reconstructing pretraining tasks used for gLMs (NTP and MLM). While a variety of approaches estimate evolutionary rate, we used PhyloP scores (Pollard et al., 2010), a widely-adopted score (Perez et al., 2025) with single-nucleotide resolution (Figure 1A). PhyloP scores quantify whether a nucleotide position is evolving slower (conserved; positive score) or more rapidly (accelerated; negative score) than predicted by a neutral evolutionary model; scores are computed from an MSA and its corresponding phylogenetic tree.

First, for CEP, the PhyloP score at the $i$-th position is predicted from the sequence context up to and including that position. Formally, given a sequence $x = x_1, x_2, ..., x_i$, the model outputs a predicted PhyloP score $\hat{y}_i$. This setup mim-

ics next token prediction (Figure 1B). Second, for MEM, the PhyloP score is predicted from bi-directional sequence context. During training, a subset of positions in the input sequence (15%) is randomly selected and masked. The model is then trained to reconstruct the PhyloP scores at **all** positions, using the entire (unmasked) portion of the sequence as context (Figure 1B). This allows the model to leverage bi-directional upstream and downstream sequence context. Because CEP and MEM operate on the same inputs used for NTP and MLM, respectively, our evolutionary prediction tasks can be combined with sequence reconstruction tasks. CEP can be combined with NTP in the autoregressive setting, and MEM can be combined with MLM in the bi-directional setting (Figure 1B).

### 3.1.1. LOSS FUNCTIONS

Gamba models predict a mean and log variance for the evolutionary rate at each position, and are trained with a Gaussian negative log likelihood (GNLL) loss:

$$= \frac{1}{N} \sum_{i=1}^{N} \left[ \frac{1}{2} \left( \log\left(\max(\sigma_i^2, \epsilon)\right) + \frac{(y_i - \mu_i)^2}{\max(\sigma_i^2, \epsilon)} \right) \right] + \text{const.}$$

where $y_i$ is the true evolutionary rate score, $\mu_i$ is the predicted mean, $\log \sigma_i^2$ is the predicted log variance, $\epsilon$ is a value used to clamp the variance for stability, and $N$ is the num-

ber of supervised positions in the batch. For the sequence prediction tasks (NTP and MLM), we used a standard cross entropy (CE) loss on the predicted sequence logits over the 4 nucleotides (A, T, C, and G), which maximizes the predicted likelihood of the true nucleotide at each position. We combined evolutionary rate prediction and sequence reconstruction by adding the two losses: $L = L_{\text{GNLL}} + L_{\text{CE}}$.

We experimented with different losses, but found little difference (Supplementary A). Thus, we report results with the standard GNLL.

### 3.2. The Gamba model architectures

The autoregressive Gamba (ArGamba) models are based on the Jamba architecture, a hybrid Transformer–Mamba model with mixture-of-experts (MoE) routing, and are roughly 66 million parameters (Lieber et al., 2024). Jamba interleaves Transformer (Vaswani et al., 2017) and Mamba (Gu & Dao, 2023) blocks to combine the global modeling capacity of attention with the efficient long-range processing of state space models. The bi-directional Gamba (Bi-Gamba) models are based on the Caduceus architecture and are roughly 4 million parameters (Schiff et al., 2024). Caduceus is a bi-directional, Mamba state space model adapted to be reverse-complement equivariant, an inductive bias that captures the double-stranded nature of DNA. For both Gamba models, we evaluated three pretraining tasks:

- **Sequence-only**: Models trained to reconstruct genomic sequences only. ArGamba, we trained models with NTP; and for Bi-Gamba, we trained models with MLM.

- **Evolution-only**: Models trained to predict PhyloP score only. For ArGamba, we trained models with CEP; and for Bi-Gamba, we trained models with MEM.

- **Sequence and evolution**: Models trained to both reconstruct sequence and predict PhyloP scores. For ArGamba, this is NTP + CEP; and for Bi-Gamba, this is MLM + MEM.

For all PhyloP-score prediction tasks, the evolutionary rate head consists of a linear layer that maps the final hidden representation at each position to the predicted mean and log variance of a Gaussian distribution over evolutionary rate scores. For sequence reconstruction tasks, a linear head maps the final hidden state to the nucleotide vocabulary.

### 3.3. Pretraining data

We trained Gamba on the human reference genome (hg38), obtained from (Basenji) (Kelley et al., 2018). We excluded repetitive and structurally ambiguous regions using Repeat-Masker annotations (Fernandes et al., 2020) and low-quality read centromere data, both obtained from the UCSC genome browser (Perez et al., 2025). This is consistent with recent gLMs such as GPN-MSA and Evo2 that address signal-to-noise ratio in the genome by adopting structured training regimes that over-sample informative regions while down-weighting repetitive regions (Benegas et al., 2024; Brixi et al., 2025). We additionally exclude repetitive and low-quality regions (more than 10% ambiguous bases ("N") within any 1000 base pair (bp) window) entirely. Chromosomes 3 and 16 are held out for validation, while chromosomes 2 and 22 are reserved for testing. After filtering, the training corpus is reduced from ∼3 billion bp to ∼1 billion bp.

For evolutionary rate prediction, we processed PhyloP scores from the Zoonomia 241-mammalian alignment (Consortium, 2020). The complete hg38 human reference genome is initialized with zeros, and PhyloP values are indexed where available; positions without a score remain zero. Across the 1,398,602,778 bp available for training, testing and validation, only 2,565,127 (0.18%) PhyloP scores were missing. All values are rounded to two decimal places.

## 4. Evaluations

We designed zero-shot evaluations to test genome language model (gLM) representations under increasing difficulty settings.

First, to assess generality and organization encoded in the representation itself, we evaluated these representations on their ability separate functional genomic regions of varying complexity and rarity without task-specific training. Next, to assess if the model recognizes true biological initiation signals from the same triplet, we tested discrimination between true methionine start codons from ATG controls that vary in coding status, whether they're in-frame or out-of-frame, and genomic proximity (near and far non-coding ATGs, same or different protein ATGs) (Table A16 and Section A). Finally, we evaluated models on existing variant effect prediction benchmarks to assess position-specific performance (Ye et al., 2025).

We compared ArGamba and Bi-Gamba to several gLMs. The Nucleotide Transformer suite provides both a human-reference-only model and a multi-species model, enabling assessment of whether evolutionary information can be captured implicitly by training on multi-species genomes and how this compares against our strategy (Dalla-Torre et al., 2025). To match ArGamba pretraining, we included Evo2, which also uses single-nucleotide tokenization and autoregressive training (Brixi et al., 2025). To match Bi-Gamba

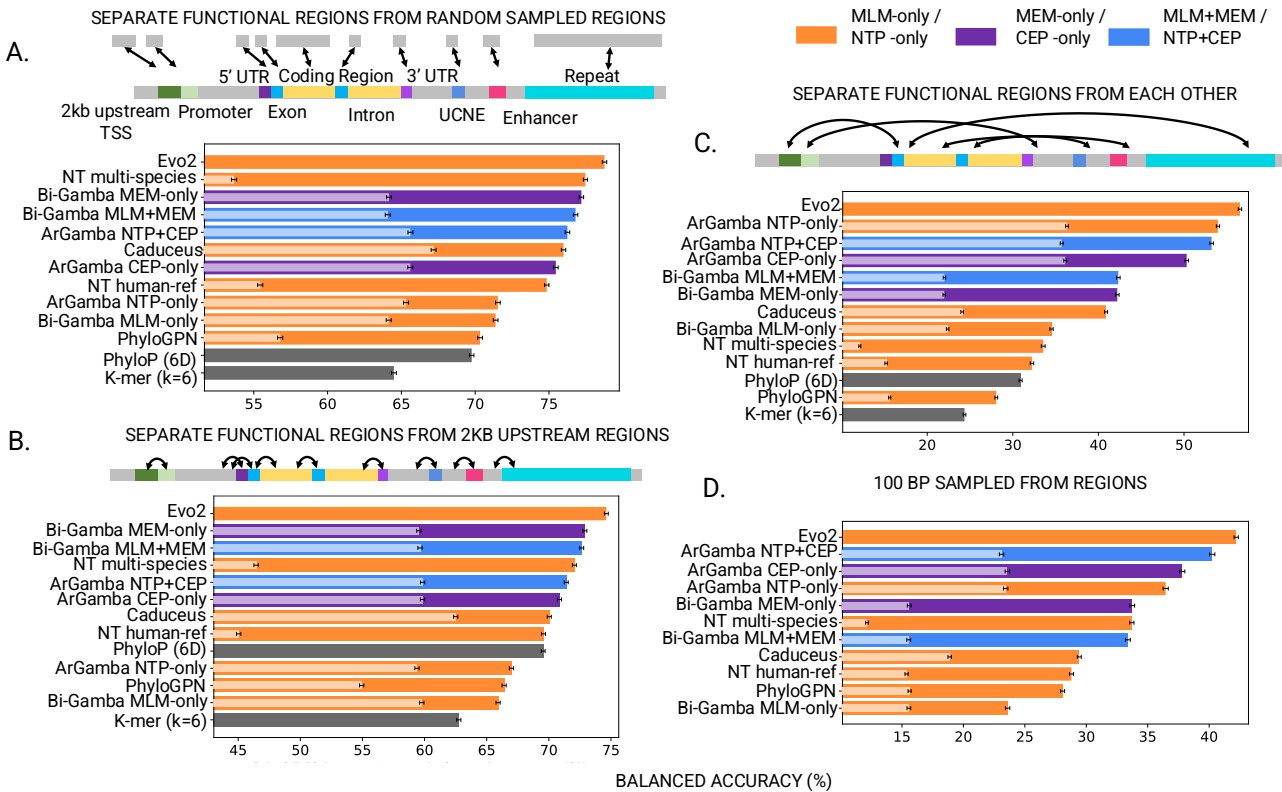

*Figure 2.* Ability of models to distinguish functional regions. A) Balanced accuracy across models for separating functional regions from randomly-sampled genomic regions. B) Balanced accuracy across models for separating functional regions from sequences directly upstream. C) Balanced accuracy across models for separating categories of regions D) Balanced accuracy across models for separating categories of regions using 100 bp subregions.

pretraining, we included Caduceus, trained on the human reference genome alone (Schiff et al., 2024). We also report PhyloGPN results, which explicitly predicts evolutionary signals from sequence and is therefore the most comparable to our evolutionary rate-based pretraining tasks (Albors et al., 2025). For model context lengths and parameter sizes see Table A1. For all evaluations we report performance on the whole human genome, as this treats Gamba more fairly with prior gLMs, which typically do not hold out human genome data. However, we note that the relative ranking of our Gamba models within each family is identical across training and held-out splits for all tasks, confirming that the benefits of evolutionary rate prediction are not dependent on the choice of split (Supplementary A2 and A3).

### 4.1. Evolutionary rate pretraining improves distinction of functional regions

As a first assessment of whether Gamba models learn biologically relevant signals in genomes, we assessed whether their representations separate functionally distinct genomic regions. Our unified evaluation includes regions of interest (ROIs) on hg38 from curated annotation sources: enhancers from VISTA (Visel et al., 2007), ultraconserved non-coding

elements (UCNEs) from UCNEbase (Dimitrieva & Bucher, 2013), repeats from RepeatMasker annotations (Fernandes et al., 2020), promoters from EPD (Périer et al., 2000), and gene-structure annotations including exons, introns, coding regions, start codons, stop codons, 2-kb upstream regions of transcription start sites (TSSs), 5′ UTRs, and 3′ UTRs from GTF- and UCSC-derived annotations (Frankish et al., 2021) (Supplementary A). ROIs were assigned mutually exclusive labels via priority-based overlap resolution.

Regions were embedded within model-specific windows that may include additional context, but each ROI was represented by mean-pooling over only its token-level representations. The ROI was centered in the input for masked language models and placed at the end of the input for autoregressive models. Representations were taken from the final hidden layer, except for Evo2, where layer 26 was used, following their analyses demonstrating that this was the most performant layer for interpretation analyses (Tables A4 to A8).

Representational separability was evaluated using leave-one-out 1-nearest-neighbor (1-NN) retrieval (Figure 2). As baselines, we report 1-NN performance for two non-gLM

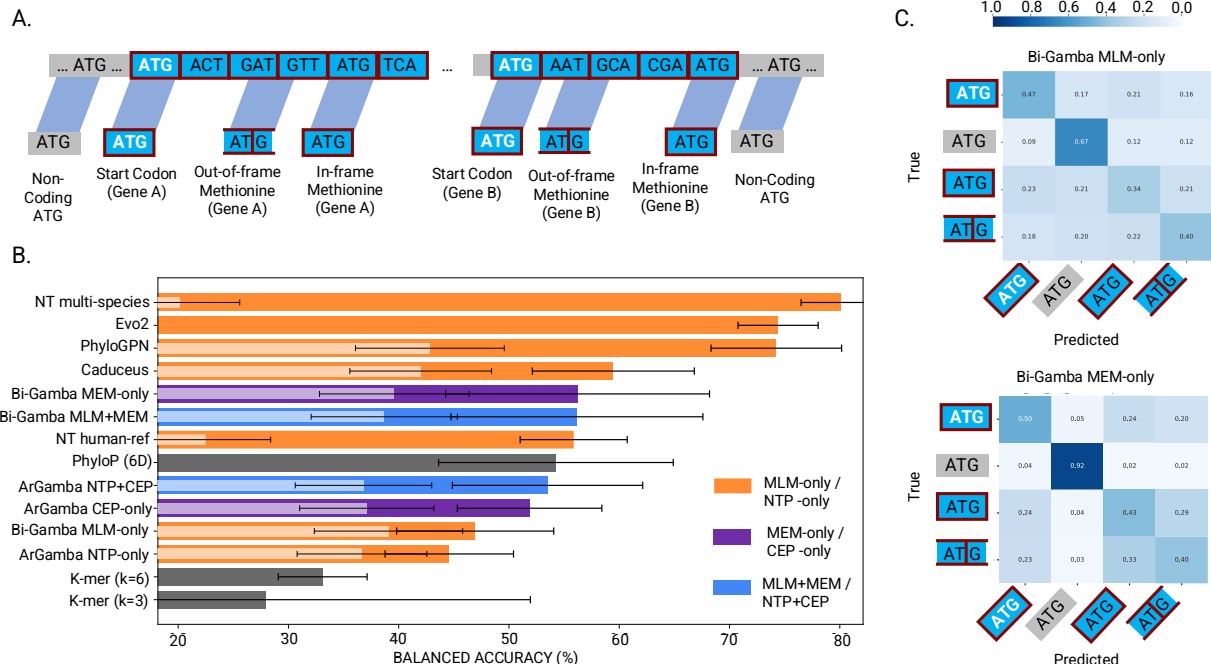

*Figure 3.* Zero-shot test of genetic code understanding via functional classification of ATG sites. A) The ATG classes. B) Balanced accuracy for leave-one-out 1-nearest-neighbor classification using sequence representations from pretrained models. Randomly-initialized models are shown in a lighter color overlay where available. C) Per-class recall heatmaps for Bi-Gamba MLM-only and MEM-only.

methods: a sequence baseline using $k$-mers ($k = 6$) and an evolutionary rate baseline summarizing PhyloP scores using six features (Supplementary Section A).

First, following most existing benchmarks, we assessed if functional regions can be distinguished from randomly sampled regions in the genome. We matched random regions such that they are length-equivalent and chromosome-matched (Figure 2A). Second, we assessed if models can separate functional regions from 2kb upstream regions of distinct function (Figure 2B). This controls for local biases in sequence composition (e.g., GC content) that make randomly sampled regions from genomes more easily distinguished compared to more spatially adjacent regions. Third, we distinguish functional regions from each other in a multi-class set-up (Figure 2C). This controls for how randomly sampled or upstream regions may bias the negative set towards more common elements (such as repeats). Fourth, we repeat the third setting, but calculate the representation as an average of a 100 bp window sampled randomly within each region, as opposed to an average of the entire region (Figure 2D). This controls for differing genomic region lengths across classes leaking information.

Across all 4 settings, Evo2 is the strongest performing model (Figure 2 A – D). However, Gamba models achieve competitive performance, with the top model achieving 95.5% to 97.7% of the performance of Evo2 despite having 0.06% to 0.95% of the parameters and outperforming all other gLMs

in 3 of 4 settings (all but distinguishing functional regions from each other). Pretraining on evolutionary rate is critical to this performance: in these 3 settings, Gamba models pretrained on evolutionary rate only or on both evolutionary rate and sequence, substantially out-perform sequence-only models. This gap is especially apparent when distinguishing functional regions from randomly sampled regions and from upstream regions, where all sequence-only Gamba variants are among the worst performing gLMs, but evolutionary rate-only or evolutionary rate-and-sequence models are among the best (Figure 2B).

As we pretrain on PhyloP scores, we also include this as a baseline to evaluate if evolutionary rate-based pretraining simply reproduces signal encoded in these scores, or if using these scores for pretraining results in deeper functional understanding. Across all settings, Gamba models outperform PhyloP6D scores, suggesting the latter. To further examine this behavior, we report per-category comparisons against PhyloP6D in Figure A5.

Although our Gamba variants suggest that including evolutionary rate in pretraining is generally superior, autoregressive sequence-only pretraining performs better than evolutionary rate-only for separating functional regions (Figure 2C). However, combining both objectives generally leads to performance comparable or better than training with either objective alone across all settings, suggesting that the benefits of the two pretraining objectives are com-

posable. We also observed additional dependencies between the downstream task setting and architecture, with ArGamba models performing best at distinguishing functional regions from each other, but Bi-Gamba models performing best at distinguishing functional regions from random or upstream regions.

Finally, we observe that contrary to existing genomic region classification benchmarks (Vishniakov et al., 2024), all gLMs outperformed their randomly initialized equivalents in our benchmark settings. This suggests zero-shot evaluations controlling for confounders are sufficiently difficult to detect the effect of pretraining. By evaluating different kinds of confounders, we additionally make the observation that some gLMs are robust to some confounders, but not others: for example, while NT multi-species performs strongly on distinguishing functional regions from randomly sampled regions, it notably performs poorly at distinguishing functional regions from each other.

### 4.2. Predicting evolution improves motif-matched functional classification of ATGs

Next, we evaluated whether incorporating evolutionary rate prediction during pretraining improves a model's ability to adapt understanding of sequences based upon context, specifically for understanding the genetic code. The region separation tasks do not test whether models can distinguish functionally distinct instances of the *same* short motif under matched sequence composition. To address this, we constructed a classification task using ATG codons in diverse functional contexts: translation start sites, in-frame and out-of-frame methionines within the same or different proteins, and non-coding ATGs at varying distances from functional start codons (Figure 3A, Section A). These contrasts require models to have learned the genetic code rather than the ATG sequence itself or simple distance-based heuristics. We evaluated all models zero-shot using leave-one-out nearest-neighbor retrieval in the representation space.

As shown in Table A16 and Figure 3B, models pretrained with evolutionary rate prediction consistently outperform their sequence-only counterparts. Within the Gamba family, our dual-trained ArGamba NTP+CEP (53.6%) substantially outperforms NTP-only (44.6%) and shows modest improvement over CEP-only (51.9%), demonstrating the complementary benefits of joint sequence and evolution modeling. This pattern holds for Bi-Gamba, where MLM+MEM and MEM-only (56.2%) outperform MLM-only (46.9%). Notably, evolutionary prediction alone provides strong gains even without sequence reconstruction, with both CEP-only and MEM-only models outperforming their sequence-only counterparts, particularly in distinguishing coding from non-coding ATGs ( Figure 3C).

Nucleotide Transformer multi-species (80.1%) achieves the

highest performance, although it is not clear if this is because it sees additional context than the other models due to its use of a 6-mer tokenization instead of nucleotide-level. However, in general, training on or being fit on multiple species appears to be effective, as evidenced by Evo2 (74.4%) and PhyloGPN (74.2%) being the next best performing models. Nucleotide Transformer human-reference and Caduceus, trained solely on human sequence, outperform our PhyloP6D baseline (54.24%), suggesting that large-scale sequence reconstruction can capture aspects of functional context within coding regions. Intriguingly, the result that evolutionary rate pretrained Gamba models do not outperform multispecies MLM/NTP models in this evaluation, despite exhibiting competitive or better performance in distinguishing coding regions in our previous benchmark suggests that evolutionary rate pretraining does not learn the codon table as well, but must instead learn some other signal to identify coding regions.

Importantly, all pretrained models substantially outperformed their random initializations (which cluster around 40% balanced accuracy), and the highly variable performance of the k=3 k-mer baseline (27.91% $\pm$ 24.03%) confirms that sequence composition is insufficient.

### 4.3. Predicting evolution improves variant effect prediction

Finally, to assess whether evolutionary rate pretraining improves predictions of the effects of fine-grained sequence changes on function, we evaluated modes on variant effect prediction (VEP) (Zhou & Troyanskaya, 2015; McLaren et al., 2016), an established benchmark that ranks genomic sequences by their pathogenicity, or their propensity to cause disease (Figure 4A). We use the benchmark proposed by Ye et al. (2025). For VEP, models that use MSAs as input, like GPN-STAR or GPN-MSA have demonstrated superior performance. We did not compare against these in our benchmark, because the aim of our study was to assess to what degree single-sequence models learn variant effects, not to maximize performance on this application.

We ranked variants in two ways (Figure 4B). First, following GPN-MSA (Benegas et al., 2024), variant effect scores are computed as the ratio of the likelihoods of the reference and mutated nucleotides, averaged across the forward and reverse strands. Second, for Gamba models trained to predict evolutionary rate, we ranked variants using the average evolutionary rate prediction (ERP) score across strands, consistent with prior work using PhyloP for VEP (Pollard et al., 2010; Benegas et al., 2024; Brixi et al., 2025; Ye et al., 2025).

Across most variant effect prediction tasks, evolutionary rate-only models consistently outperformed sequence-only models (Tables A9 and A11 to A14). Sequence likelihood

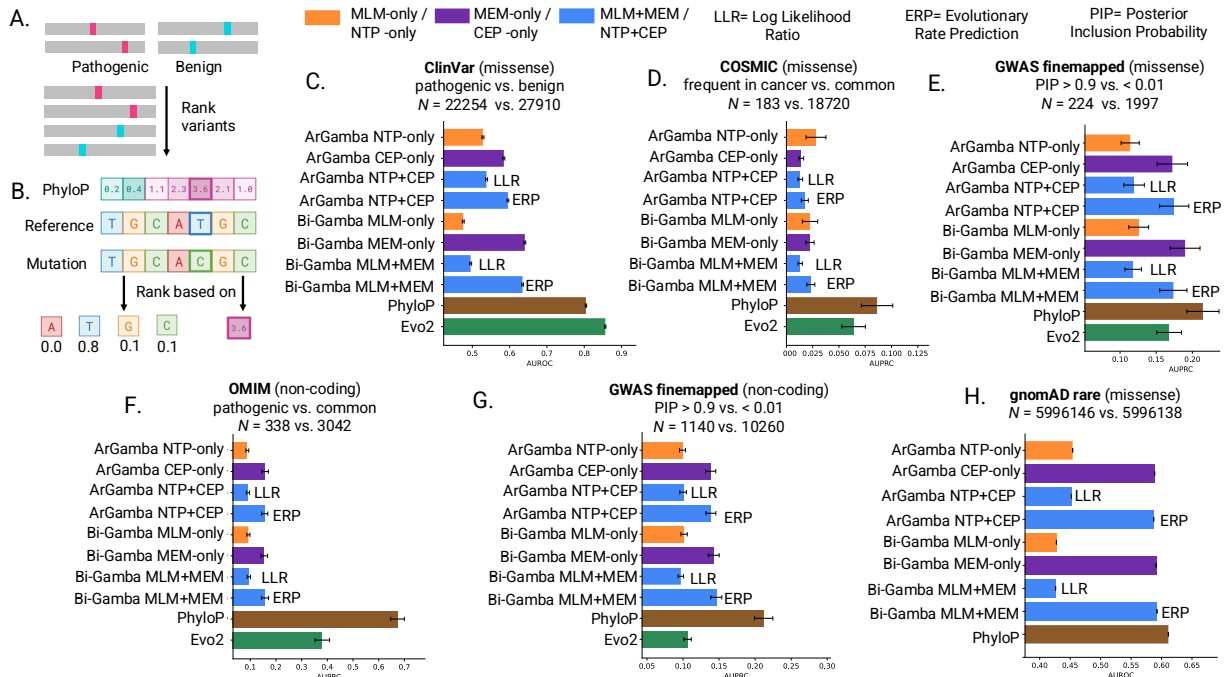

Figure 4. Variant effect prediction (VEP). A) The VEP task. B) Scoring and ranking variants. C)-H) Performance on various VEP tasks. Evo2 results not available for H due to evaluation size.

alone provides weak and inconsistent signal for pathogenicity, with ArGamba NTP-only and Bi-Gamba MLM-only often performing near random, particularly on non-coding and rare-variant benchmarks. In contrast, predicted evolutionary rate scores from CEP- or MEM-pretrained models yielded substantially stronger performance.

Joint pretraining on sequence and evolutionary rate does not always outperform single-objective training, and its effect depends on both the model family and the evaluation. For ArGamba, joint training (NTP+CEP) typically improves sequence log-likelihood relative to NTP-only across most tasks, but yields mixed results when evaluated using predicted evolutionary rate, with CEP-only outperforming the joint model on several non-coding benchmarks (Tables A9 and A11 to A14). In contrast, for Bi-Gamba, dual training (MLM+MEM) more consistently improves predicted evolutionary rate relative to MEM-only across the majority of tasks, but often degrades sequence log-likelihood compared to MLM-only on missense benchmarks, with the exception of ClinVar (Tables A10 and A11). As a result, the strongest model varies by task: Bi-Gamba models tend to score best on missense variants, while ArGamba models are competitive or superior on several non-coding tasks.

Pretraining models to predict evolutionary rate can lead to strong zero-shot performance on tasks that sequence reconstruction models perform poorly. Across all VEP tasks, true PhyloP scores remain the strongest single predictor. Evo2

(7B parameters) also achieves strong likelihood-based VEP performance, particularly on ClinVar (Table A9). Despite being orders of magnitude smaller, ArGamba (66M parameters) and Bi-Gamba (4M parameters) match or outperform Evo2 on several tasks when evaluated using predicted evolutionary rate, including GWAS fine-mapped missense and non-coding variants (Tables A11 and A14).

## 5. Conclusion

We demonstrated that predicting evolutionary rates from the nucleotide sequence improves the zero-shot performance of genome language model representations in the human genome. We introduced two evolutionary rate-based pretraining tasks, used them to train the Gamba family of models, and demonstrated their advantages over sequence reconstruction tasks. We furthermore developed two novel zero-shot evaluations for gLMs. Evolutionary rate-only models outperformed sequence-only baselines on several benchmarks, underscoring the effectiveness of evolutionary rate as a training target.

These findings point towards pretraining strategies that more closely align model learning with the underlying biology of the genome. While we use PhyloP scores derived from 241 mammalian species from the Zoonomia project (Consortium, 2020), future work could expand to alternative scores, including PhastCons scores (Zhou et al., 2011), which capture evolutionary rates over broader contexts, and

PhyloP scores from alternative MSAs, such as vertebrate alignments, which have been shown to outperform the 241-mammalian–derived PhyloP scores on variant effect prediction (Benegas et al., 2024). We note that we chose PhyloP scores over PhastCons for two reasons. One, its single-base resolution matches the information density of sequence reconstruction tasks exactly, allowing us to compose our prediction task with the single-sequence pretraining tasks most gLMs are trained with. Two, it captures conservation and acceleration signals, whereas PhastCons are biased toward detecting conserved elements. However, both scores discard information relative to the full MSA. For example, PhyloP scores don't handle indels, the rate heterogeneity over the evolutionary tree, or asymmetry in the rate of evolutionary depending on direction of substitution. In addition, relative to MSAs, it is lossy of information like the distribution over nucleotides at a position.

To isolate the effect of evolutionary rate as a pretraining signal, we used smaller models with shorter context windows and less training data than current SOTA gLMs. Therefore, the Gamba models we present are likely underparametrized, and future work should investigate how these findings interact with increased model parameters and data scaling.

In sum, we demonstrate the potential of combining comparative genomics and deep learning to understand genomes. Despite being a fundamental strategy in bioinformatics methods used to understand genomes, comparative genomics has not yet been thoroughly investigated for training gLMs. By demonstrating that predicting evolutionary rate can be composed with sequence reconstruction strategies currently employed to train gLMs, we hope to spur future investigation in incorporating evolutionary signal into gLMs.

## 6. Resource Availability

Code and model are made available on Github and Hugging-Face respectively.

## Impact Statement

This paper presents work whose goal is to advance the field of machine learning. There are many potential societal consequences of our work, none of which we feel must be specifically highlighted here.

COMPETING INTEREST STATEMENT

B.W. is currently employed as the Senior Vice President and Head of Biomedical AI at Xaira Therapeutics. He also serves as a scientific advisor to Deep Genomics, Shift Biosciences, and VieCure Inc.

ACKNOWLEDGMENTS

We thank Vector Institute compute infrastructure and Canada Research Chairs support for AM and MEC. MEC was also funded by the NSERC CGS-D. We thank David Knowles, members of the Moses lab, and Michael Brudno for discussions on this project. We thank Katherine Pollard and Adam Siepel for their help in working with the PhyloP scores.

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

# A. Appendix

## Genomic Region Selection

We assembled a diverse set of genomic categories from established resources to evaluate model performance across simple, complex, and rare regulatory regions.

Promoter is an essential region in the non-coding genome that is important for transcription initiation. Promoter annotations were obtained from the Eukaryotic Promoter Database (EPD) (Périer et al., 2000).

Enhancers are non-coding regulatory regions that control tissue-specific or cell-specific gene expression (Visel et al., 2007). Enhancers were sourced from the VISTA Enhancer Browser (Visel et al., 2007), which experimentally validate putative enhancers in developing mouse embryos by testing whether they drive gene expression in specific tissues.

Ultra-conserved non-coding elements (UCNEs) were taken from UCNEbase (Dimitrieva & Bucher, 2013), and are defined as sets of non-coding regions in the genome longer than 200 bp with $\geq$95% sequence identity between humans and chickens.

Repeats encompass a broad class of genomic regions, short or long, that appear multiple times in the genome, or otherwise are considered low complexity DNA regions (Fernandes et al., 2020). In some cases, repeats are caused by transposable regions, mobile genetic regions that can move themselves and duplicate in the genome despite being largely non-functional for the host species. Repeats were extracted from the UCSC RepeatMasker track (Fernandes et al., 2020).

Protein-coding exons, introns, upstream transcription start site (2kb upstream of TSS), 5'UTRs, 3'UTRs, and coding regions were derived from the GENCODE human genome annotations (GTF format) (Frankish et al., 2021). All regions from the GENCODE human genome annotations were filtered to canonical transcripts: a single, representative transcript identified at every gene (usually with the highest coverage of conserved exons, highest expression, longest coding sequence, etc.) (Dyer et al., 2025). Introns are regions within genes that are spliced out, and do not encode for proteins, but can play important roles in mRNA processing. Exons are regions within genes that remain after splicing, but can be non-coding, meaning they do not get translated to proteins, if they are UTRs. 5'UTRs and 3'UTRs are specific classes of exons involved in splicing and regulation of translation as well as RNA stability. The region defined as 2kb upstream of the Transcription Start Site (TSS) is considered a functionally rich region which may contain sequences involved in transcription initiation, including promoters.

## Loss Experiments

We experimented with a focal Gaussian Negative Likelihood Loss (GNLL), which upweights nucleotides with high residuals to address the rarity of high magnitude PhyloP scores (Lin et al., 2017). Both losses produced nearly identical sequence reconstruction losses (Figure A1A), and did not consistently improve predictions (Figure A2). Therefore, in the interest of demonstrating that our results held true in the most naive case, we report GNLL.

We weighed both components of the loss evenly: we found that empirically, this led to good convergence of the model (Figure A3). ArGamba NTP+CEP has a converged CE loss of 1.303, compared to 1.289 for NTP-only. For Bi-Gamba, MLM+MEM has a CE loss of 1.180 and MLM-only of 1.176. We verified that these losses were not significantly different from other pretrained gLMs on our validation dataset: Caduceus achieves a MLM CE loss of 1.178.

We further experimented with normalizing PhyloP scores to improve loss on evolutionary rate reconstruction. However, Gaussian normalization of the PhyloP scores did not provide notable improvements in training stability, and even resulted in less-accurate PhyloP score predictions. With Bi-Gamba models at 190k steps on held-out data, the unnormalized model achieves Pearson r=0.727 and Spearman $\rho$=0.689, while the Gaussian-normalized variant performs slightly worse (Pearson r=0.716, Spearman $\rho$=0.679), with a shallower regression slope (0.45 vs 0.50), we therefore retained the raw scale (Figure A4).

## Dataset Construction

All functional regions of interest (ROIs) were processed under a common interval convention (0-based, half-open) and filtered to enforce mutual exclusivity across categories. When regions from different categories overlapped, a fixed priority ordering was applied, ensuring that each genomic locus was assigned to at most one functional class. This yielded functional ROIs with exclusive labels and no overlap at the ROI level.

For each retained functional ROI, we required the existence of a strand-aware proximal upstream region of identical length, located 2 kb upstream of the ROI. Upstream regions were required not to overlap any ROI from the same functional category. ROIs without a valid upstream counterpart were excluded. Each ROI–upstream pair was assigned a shared identifier.

Using this canonical set of functional ROIs, we derived two types of length-matched random regions: (1) non-annotated random regions, sampled from genomic loci that did not overlap any known functional annotation across all categories, and (2) category-matched random regions, sampled from genomic loci that did not overlap the same functional category as the ROI, but could overlap other annotations. Random regions were sampled on the same chromosome and matched exactly in length to their corresponding ROI.

This process yielded a dataset consisting of functional ROIs with exclusive labels, paired proximal upstream regions, length-matched non-annotated random regions, and length-matched category-matched random regions. All evaluation tasks were derived from this dataset.

**Embedding and Pooling**

For each region (functional ROI, upstream region, or random region), we extracted a model-specific genomic context window prior to embedding. Context window size and positioning followed the conventions of each model family: asymmetric context for causal models (e.g., upstream-anchored), symmetric context for bi-directional models, and fixed window sizes per model (e.g., 2048 bp for the Gamba models, 481 bp for phyloGPN).

Each region was embedded using the corresponding genomic language model. Token-level representations were then pooled by masked averaging over the region of interest only, excluding padding tokens. This yielded a single vector representation per region.

For the multiclass classification task only, additional length control was applied by sampling fixed-length sub-regions from within each ROI.

All evaluations were performed using leave-one-out 1-nearest-neighbor (1-NN) retrieval to assess representational separability without training task-specific classifiers.

**Evaluation Tasks**

**Task 1: Functional ROI vs Category-Matched Random Genome**    We tested whether representations distinguished functional regions from generic genomic regions that were not part of the same functional category. Functional ROIs were paired with length-matched random regions sampled from the same chromosome that did not overlap the ROI's category but could overlap other annotations.

**Task 2: Functional ROI vs Proximal Upstream Region**    We tested whether representations localized functional signal relative to immediately flanking genomic sequence. Each functional ROI was paired with a strand-aware upstream region of identical length located 2 kb upstream. Upstream regions shared local genomic context, base composition, and neighborhood effects with the ROI, making this a stringent test of functional localization rather than global sequence properties.

**Task 3: Multiclass Functional Region Classification**    We evaluated whether representations distinguished among distinct functional categories. Functional ROIs with mutually exclusive labels were used.

**Task 4: Multiclass Functional Region Classification (100bp Sample)**    This task is the same as task 3 with the exception that 100bp of the regions are sampled to remove region length as a confound. In this task fixed-length sub-regions (of 100 bp) were sampled uniformly from within each ROI. Only these sub-regions were pooled and compared.

By constructing a single canonical dataset and deriving multiple evaluation tasks spanning coarse to fine resolution, this benchmark systematically probed functional signal, class specificity, and spatial localization in genomic language model representations.

**ArGamba & Bi-Gamba Training**

Models were trained for roughly seven epochs on either a single NVIDIA L40S (48GB) or NVIDIA RTX A6000 (48GB). We used the Adam optimizer with a learning rate schedule defined by a linear warmup followed by inverse square root

decay, implemented via a LambdaLR scheduler.

## Variant Effect Prediction

We evaluate variant effect prediction (VEP) using a unified, single-task evaluation pipeline applied consistently across multiple benchmark datasets from the SongLab GPN-Star suite hosted on Hugging Face (Ye et al., 2025). Specifically, we use the test splits of the following datasets: songlab/clinvar_vs_benign, songlab/cosmic, songlab/ukb_finemapped_coding, songlab/omim_traitgym, songlab/gnomad_balanced, and songlab/ukb_finemapped_nc_traitgym. All datasets are evaluated in a strictly zero-shot setting, without any task-specific fine-tuning.

For each task variant, we extract a 2048 bp reference-centered sequence window from the hg38 genome assembly. We apply strict preprocessing and quality control to ensure consistency across tasks: only single-nucleotide variants with A/C/G/T alleles are retained; the extracted sequence window must contain only canonical bases; and the reference allele must exactly match the genome at the variant position. Variants failing any criterion are excluded, and detailed counters are maintained to track drop reasons. This filtering ensures that all evaluated variants are compatible with the model input assumptions and genome reference.

Model predictions are computed on both the forward strand and the reverse complement of each sequence, and final scores are obtained by averaging the two, enforcing strand invariance. For sequence-based models, we compute a per-variant log-likelihood ratio (LLR), defined as the difference between the log-likelihood of the alternate allele and that of the reference allele. For models predicting evolutionary rate, we use the scalar evolutionary rate prediction at the variant position, the Evolutionary Rate Prediction (ERP). For models trained jointly on sequence and evolutionary rate objectives, both LLR-based and ERP-based scores are evaluated.

Performance is measured using AUROC or AUPRC, depending on the task specification provided by the benchmark. Statistical uncertainty is estimated using nonparametric bootstrapping with 200 resamples, applied after filtering to the final set of valid variants. Metrics are only computed when both classes are present after filtering.

As baselines, we report PhyloP scores extracted at the variant position from a multi-species track, and, where available, Evo2 predictions loaded from row-aligned parquet files distributed alongside the corresponding Hugging Face datasets (e.g., under predictions/Evo2_7B.parquet). Evo2 scores are explicitly aligned to the exact subset of variants retained after preprocessing to ensure that all models and baselines are evaluated on identical variant sets. For each experiment, we report the evaluation metric, its bootstrap standard error, and the number of variants used.

## ATG-separation task

We constructed a classification task to evaluate whether genomic language models can distinguish ATG codons in functionally distinct contexts. All ATG regions of interest share identical sequence (ATG) but differ in their position relative to protein-coding structures and regulatory regions, with the exception of models with non-overlapping 6-mer tokenization (Nucleotide Transformer models), the ATG representation includes the 6-mer token(s) spanning the ATG codon, which may contain up to 3bp of flanking sequence. Other models pool single-nucleotide or codon-level representations of the ATG itself. We extracted coding sequence (CDS) annotations from GENCODE v44 (Frankish et al., 2021) filtered to MANE Select v1.3 canonical transcripts (Morales et al., 2022), representing the single most biologically relevant isoform per protein-coding gene. For each transcript, we reconstructed the complete CDS by extracting all CDS features from the GTF, applying GTF phase trimming to the first segment to ensure correct reading frame alignment, concatenating segments in transcript direction (5' to 3'), and reverse-complementing for minus-strand transcripts. We retained only transcripts with valid ATG start codons and CDS length $\geq 3$ bp, yielding 19,370 canonical transcripts across chromosomes 1–22.

For each reference transcript, we identified ATG contexts relative to the translation start position (anchor): (1) the translation start itself (CDS offset 0), (2) a non-coding ATG 2–5 kb away (outside any CDS), (3) a non-coding ATG >100 kb away, (4) an in-frame methionine within the same protein (CDS offset $o > 0$ where $o \bmod 3 = 0$), (5) an out-of-frame ATG within the same protein ($o \bmod 3 \neq 0$). We generated labels for chromosomes 1–22 and retained only examples where all labels could be assigned. To ensure consistency across models, we performed chromosome-stratified sampling to balance representation (2,000 start codon examples evenly distributed across chromosomes 1–22 where possible).

We evaluated all models using leave-one-out 1-nearest neighbor classification in representation space. For each model, we extracted a context size as described for the zero-shot representation task and computed the mean representation over the 3 bp ATG window, before identifying the nearest neighbor among the remaining 9,999 examples using cosine distance, and

computed balanced accuracy.

**Bioinformatics Baselines**

Baselines include a k-mer model (k=6, $L_2$-normalized 6-mer frequency vectors), a k-mer model (k=3, $L_2$-normalized 3-mer frequency vectors), and PhyloP (6D summary of PhyloP scores: mean, standard deviation, fraction positive/negative, mean of positive/negative scores).

**Training Data**

Chromosome sizes and centromere annotations were downloaded from the UCSC Genome Browser (Perez et al., 2025). The full human genome sequence (hg38.ml.fa) was obtained from the Basenji Barnyard resource (Kelley et al., 2018), and repeat regions were collected from the UCSC RepeatMasker track (Fernandes et al., 2020). PhyloP scores were downloaded from the 241-mammalian alignment hub provided by the Comparative Genomics Lab at UCSC (Pollard et al., 2010). Chromosomes 2 and 22 were held out for test, and chromosomes 16 and 3 for validation.

**Supplementary Figures & Tables**

*Table A1.* All models evaluated with parameter counts (in millions) and context lengths (in 1,000 bp).

| Model | Params (M) | Context (kbp) |
|---|---|---|
| ARGAMBA NTP-ONLY | 66.5 | 2 |
| ARGAMBA CEP-ONLY | 66.5 | 2 |
| ARGAMBA NTP+CEP | 66.5 | 2 |
| BI-GAMBA NTP-ONLY | 3.9 | 2 |
| BI-GAMBA MEM-ONLY | 3.9 | 2 |
| BI-GAMBA MLM+MEM | 3.9 | 2 |
| Evo2 | 7000 | 2 |
| NT multi-species | 498.3 | 6 |
| NT human-ref | 480.4 | 6 |
| PhyloGPN | 83.2 | 0.481 |
| HyenaDNA | 6.6 | 160 |
| Caduceus | 7.7 | 131 |
| K-mer (k=6) | 0.0 | 2 |
| PhyloP (6D) | 0.0 | 2 |

*Table A2.* Held-out vs. training performance on ROI classification ($\Delta = $ test $-$ training, balanced accuracy %).

| Model | Random | Upstream | Rnd-noannot | MC-ROI | MC-ROI100bp |
|---|---|---|---|---|---|
| ARGAMBA NTP-ONLY | -0.70 | -2.36 | -1.09 | -1.19 | +0.48 |
| ARGAMBA CEP-ONLY | -1.42 | -0.25 | -1.05 | -1.00 | -0.54 |
| ARGAMBA NTP+CEP | -0.92 | -1.43 | -1.19 | -0.91 | -1.12 |
| BI-GAMBA MLM-ONLY | -1.32 | -1.38 | -0.80 | -1.93 | -0.78 |
| BI-GAMBA MEM-ONLY | -1.30 | -1.18 | -0.70 | -1.05 | -0.41 |
| BI-GAMBA MLM+MEM | -1.17 | -0.77 | -0.06 | -2.39 | -1.38 |

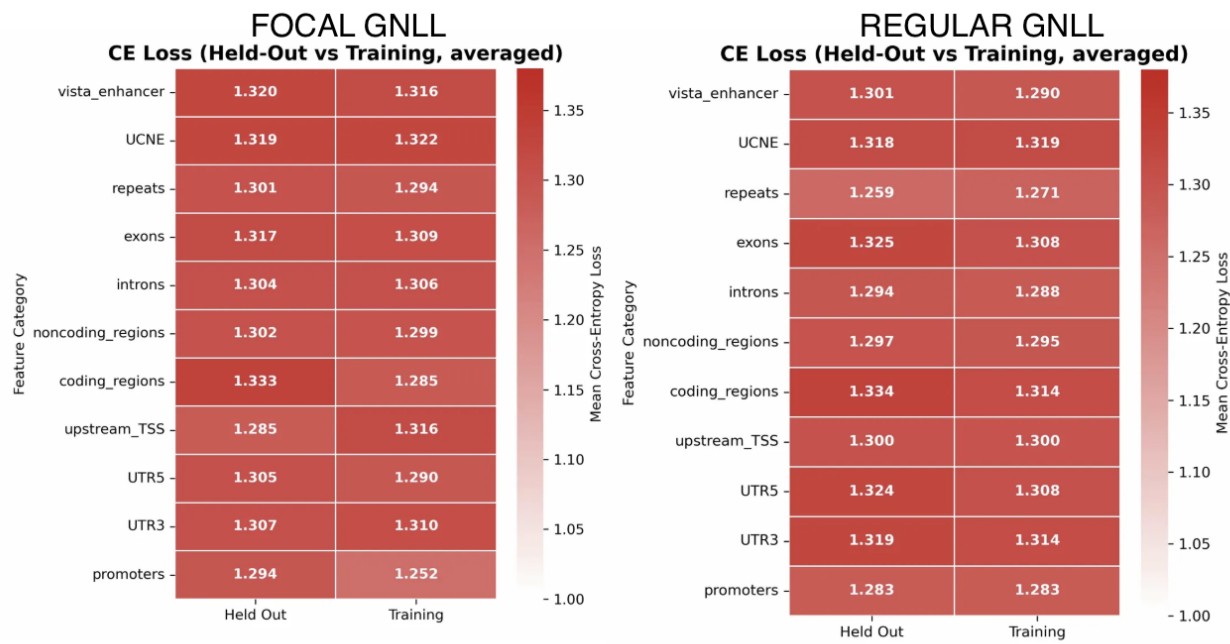

*Figure A1.* Region-level CE losses from model trained with focal and non-focal GNLL on PhyloP scores.

*Table A3.* Held-out chromosome evaluation across VEP tasks. $\Delta$ = Held-out $-$ Full.

| Family | Training task | VEP task | Full | Held-out | $\Delta$ |
|---|---|---|---|---|---|
| ArGamba | NTP-only | ClinVar pathogenic vs benign missense | 0.528 | 0.527 | -0.001 |
| ArGamba | CEP-only | ClinVar pathogenic vs benign missense | 0.585 | 0.568 | -0.017 |
| ArGamba | NTP+CEP | ClinVar pathogenic vs benign missense | 0.595 | 0.577 | -0.019 |
| Bi-Gamba | MLM-only | ClinVar pathogenic vs benign missense | 0.476 | 0.480 | +0.004 |
| Bi-Gamba | MEM-only | ClinVar pathogenic vs benign missense | 0.641 | 0.618 | -0.023 |
| Bi-Gamba | MLM+MEM | ClinVar pathogenic vs benign missense | 0.635 | 0.612 | -0.023 |
| ArGamba | NTP-only | COSMIC frequent in cancer vs common missense | 0.028 | 0.147 | +0.119 |
| ArGamba | CEP-only | COSMIC frequent in cancer vs common missense | 0.014 | 0.020 | +0.006 |
| ArGamba | NTP+CEP | COSMIC frequent in cancer vs common missense | 0.017 | 0.046 | +0.029 |
| Bi-Gamba | MLM-only | COSMIC frequent in cancer vs common missense | 0.022 | 0.089 | +0.067 |
| Bi-Gamba | MEM-only | COSMIC frequent in cancer vs common missense | 0.022 | 0.025 | +0.003 |
| Bi-Gamba | MLM+MEM | COSMIC frequent in cancer vs common missense | 0.023 | 0.035 | +0.012 |
| ArGamba | NTP-only | GWAS finemapped missense causal vs matched | 0.114 | 0.093 | -0.021 |
| ArGamba | CEP-only | GWAS finemapped missense causal vs matched | 0.172 | 0.140 | -0.032 |
| ArGamba | NTP+CEP | GWAS finemapped missense causal vs matched | 0.175 | 0.153 | -0.022 |
| Bi-Gamba | MLM-only | GWAS finemapped missense causal vs matched | 0.126 | 0.143 | +0.017 |
| Bi-Gamba | MEM-only | GWAS finemapped missense causal vs matched | 0.190 | 0.167 | -0.023 |
| Bi-Gamba | MLM+MEM | GWAS finemapped missense causal vs matched | 0.174 | 0.150 | -0.024 |
| ArGamba | NTP-only | OMIM noncoding pathogenic vs common | 0.089 | 0.097 | +0.009 |
| ArGamba | CEP-only | OMIM noncoding pathogenic vs common | 0.158 | 0.217 | +0.059 |
| ArGamba | NTP+CEP | OMIM noncoding pathogenic vs common | 0.156 | 0.223 | +0.066 |
| Bi-Gamba | MLM-only | OMIM noncoding pathogenic vs common | 0.093 | 0.095 | +0.003 |
| Bi-Gamba | MEM-only | OMIM noncoding pathogenic vs common | 0.155 | 0.167 | +0.012 |
| Bi-Gamba | MLM+MEM | OMIM noncoding pathogenic vs common | 0.157 | 0.188 | +0.030 |
| ArGamba | NTP-only | GWAS finemapped | 0.099 | 0.106 | +0.006 |
| ArGamba | CEP-only | GWAS finemapped | 0.138 | 0.144 | +0.006 |
| ArGamba | NTP+CEP | GWAS finemapped | 0.139 | 0.139 | +0.001 |
| Bi-Gamba | MLM-only | GWAS finemapped | 0.101 | 0.104 | +0.003 |
| Bi-Gamba | MEM-only | GWAS finemapped | 0.143 | 0.150 | +0.008 |
| Bi-Gamba | MLM+MEM | GWAS finemapped | 0.146 | 0.151 | +0.005 |

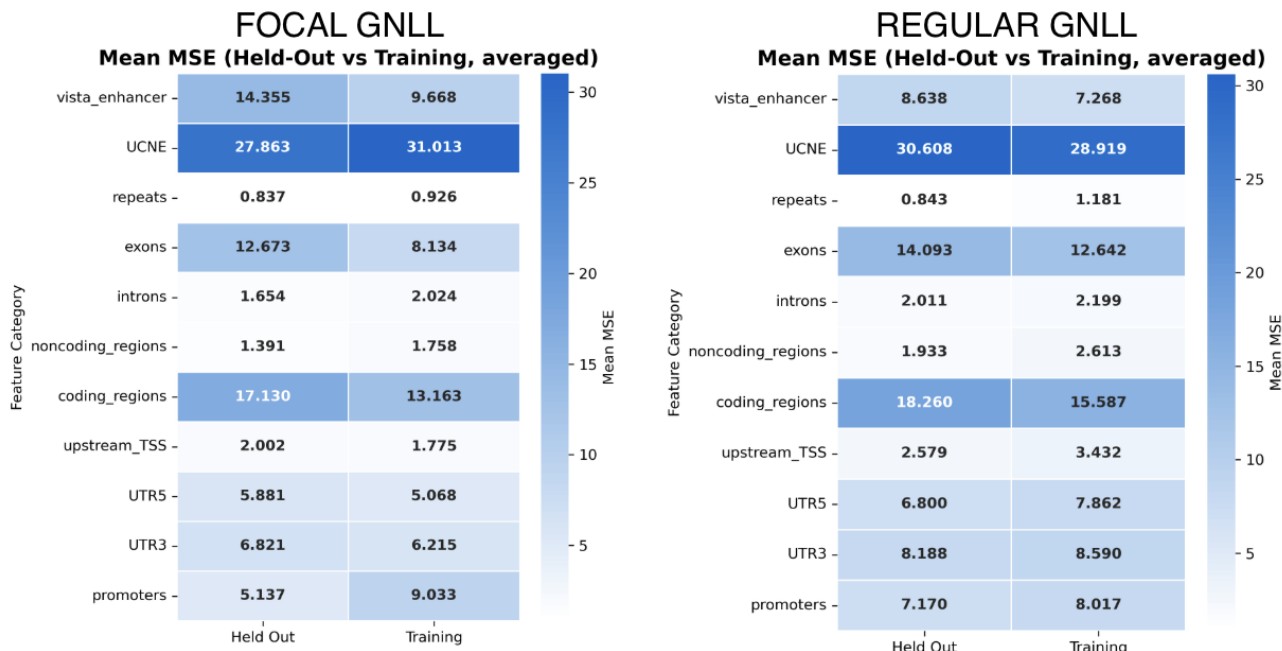

*Figure A2.* Region-level MSE calculated from predicted mean of the focal and non-focal GNLL on PhyloP scores.

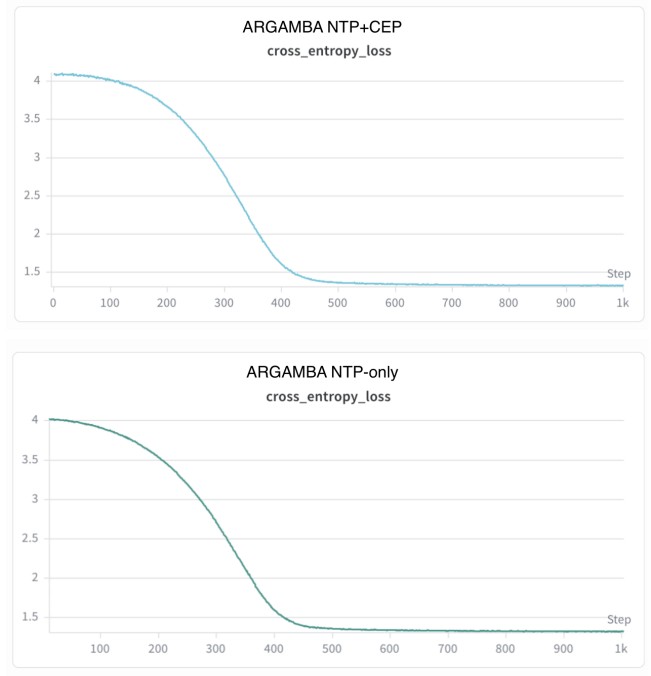

*Figure A3.* Loss curves for cross-entropy loss of the ArGamba models trained using sequence reconstruction only (ArGamba NTP-only) and sequence reconstruction + evolutionary rate prediction (ArGamba NTP+CEP).

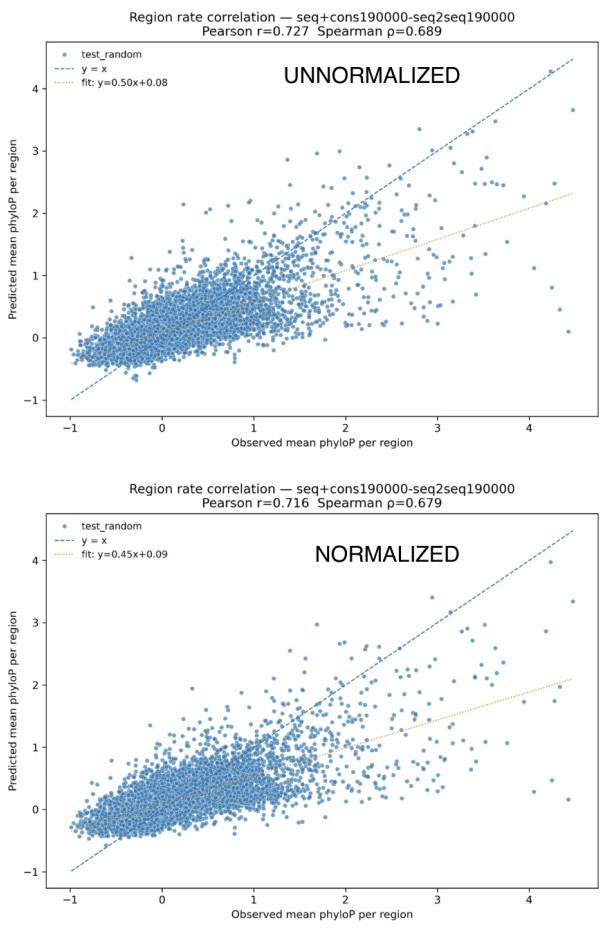

*Figure A4.* Correlation between held out genomic regions true and predicted PhyloP scores for normalized and unnormalized PhyloP score training.

*Table A4.* Multiclass functional region classification performance (balanced accuracy, %). Random initialization corresponds to step 0; pretrained corresponds to step 44k for Gamba models. Reported values are mean $\pm$ standard error.

| Model | Random Init (%) | Pretrained (%) | $\triangle$ BA |
|---|---|---|---|
| ARGAMBA NTP-ONLY | $36.34 \pm 0.18$ | $\mathbf{53.97 \pm 0.19}$ | +17.63 |
| ARGAMBA CEP-ONLY | $36.13 \pm 0.17$ | $50.31 \pm 0.20$ | +14.18 |
| ARGAMBA NTP+CEP | $35.71 \pm 0.17$ | $53.23 \pm 0.20$ | +17.51 |
| BI-GAMBA MLM-ONLY | $22.39 \pm 0.15$ | $34.53 \pm 0.19$ | +12.14 |
| BI-GAMBA MEM-ONLY | $21.99 \pm 0.14$ | $42.21 \pm 0.21$ | +20.22 |
| BI-GAMBA MLM+MEM | $22.02 \pm 0.15$ | $42.33 \pm 0.21$ | +20.31 |
| CADUCEUS | $24.09 \pm 0.15$ | $40.90 \pm 0.18$ | +16.81 |
| HYENADNA | $38.59 \pm 0.17$ | $34.40 \pm 0.14$ | −4.19 |
| PHYLOGPN | $15.62 \pm 0.13$ | $28.07 \pm 0.16$ | +12.45 |
| NUCLEOTIDE TRANSFORMER (HUMAN) | $15.20 \pm 0.14$ | $32.24 \pm 0.21$ | +17.04 |
| NUCLEOTIDE TRANSFORMER (MULTI-SPECIES) | $12.13 \pm 0.12$ | $33.55 \pm 0.22$ | +21.42 |
| EVO2 | – | $\mathbf{56.54 \pm 0.20}$ | – |
| K-mer model ($k = 6$) | – | $24.36 \pm 0.16$ | – |
| PhyloP (6D) | – | $30.91 \pm 0.18$ | – |

*Table A5.* Multiclass functional region classification using fixed 100 bp sub-regions (balanced accuracy, %). Random initialization corresponds to step 0; pretrained corresponds to step 44k for Gamba models. Reported values are mean $\pm$ standard error.

| Model | Random Init (%) | Pretrained (%) | $\Delta$ BA |
|---|---|---|---|
| ARGAMBA NTP-ONLY | $23.44 \pm 0.18$ | $36.46 \pm 0.21$ | +13.02 |
| ARGAMBA CEP-ONLY | $23.58 \pm 0.19$ | $37.81 \pm 0.21$ | +14.23 |
| ARGAMBA NTP+CEP | $23.11 \pm 0.18$ | $\mathbf{40.25 \pm 0.22}$ | +17.14 |
| BI-GAMBA MLM-ONLY | $15.56 \pm 0.15$ | $23.60 \pm 0.18$ | +8.04 |
| BI-GAMBA MEM-ONLY | $15.59 \pm 0.15$ | $33.74 \pm 0.20$ | +18.15 |
| BI-GAMBA MLM+MEM | $15.55 \pm 0.16$ | $33.38 \pm 0.20$ | +17.83 |
| CADUCEUS | $18.89 \pm 0.14$ | $29.43 \pm 0.16$ | +10.54 |
| HYENADNA | $31.85 \pm 0.14$ | $31.53 \pm 0.14$ | $-0.32$ |
| PHYLOGPN | $15.62 \pm 0.13$ | $28.07 \pm 0.16$ | +12.45 |
| NUCLEOTIDE TRANSFORMER (HU-MAN) | $15.38 \pm 0.13$ | $28.79 \pm 0.16$ | +13.41 |
| NUCLEOTIDE TRANSFORMER (MULTI-SPECIES) | $12.16 \pm 0.12$ | $33.71 \pm 0.17$ | +21.54 |
| EVO2 | – | $\mathbf{42.20 \pm 0.17}$ | – |

*Table A6.* Binary classification of functional ROIs versus category-matched random genomic regions (balanced accuracy, %). Values are mean $\pm$ SE.

| Model | Random Init (%) | Pretrained (%) |
|---|---|---|
| ARGAMBA NTP-ONLY | $65.32 \pm 0.17$ | $71.54 \pm 0.16$ |
| ARGAMBA CEP-ONLY | $65.60 \pm 0.17$ | $75.47 \pm 0.15$ |
| ARGAMBA NTP+CEP | $65.61 \pm 0.17$ | $76.25 \pm 0.15$ |
| BI-GAMBA MLM-ONLY | $64.14 \pm 0.17$ | $71.39 \pm 0.16$ |
| BI-GAMBA MEM-ONLY | $64.16 \pm 0.17$ | $77.20 \pm 0.15$ |
| BI-GAMBA MLM+MEM | $64.08 \pm 0.17$ | $76.80 \pm 0.15$ |
| CADUCEUS | $67.18 \pm 0.17$ | $75.99 \pm 0.15$ |
| HYENADNA | $64.87 \pm 0.17$ | $63.47 \pm 0.17$ |
| PHYLOGPN | $56.78 \pm 0.18$ | $70.33 \pm 0.16$ |
| NUCLEOTIDE TRANSFORMER (HU-MAN) | $55.43 \pm 0.18$ | $74.85 \pm 0.15$ |
| NUCLEOTIDE TRANSFORMER (MULTI-SPECIES) | $53.66 \pm 0.18$ | $77.47 \pm 0.14$ |
| EVO2 | – | $\mathbf{78.76 \pm 0.15}$ |
| K-mer model ($k = 6$) | – | $64.49 \pm 0.17$ |
| PhyloP (6D) | – | $69.77 \pm 0.15$ |

*Table A7.* Binary classification of functional ROIs versus non-annotated genomic background (balanced accuracy, %).

| Model | Random Init (%) | Pretrained (%) |
|---|---|---|
| ARGAMBA NTP-ONLY | 65.52 ± 0.17 | 71.83 ± 0.16 |
| ARGAMBA CEP-ONLY | 65.82 ± 0.17 | 75.82 ± 0.15 |
| ARGAMBA NTP+CEP | 65.68 ± 0.17 | 76.79 ± 0.15 |
| BI-GAMBA MLM-ONLY | 64.59 ± 0.17 | 71.76 ± 0.16 |
| BI-GAMBA MEM-ONLY | 64.41 ± 0.17 | 77.56 ± 0.14 |
| BI-GAMBA MLM+MEM | 64.39 ± 0.17 | 77.19 ± 0.15 |
| CADUCEUS | 67.25 ± 0.17 | 76.06 ± 0.15 |
| HYENADNA | 65.03 ± 0.17 | 63.56 ± 0.17 |
| PHYLOGPN | 56.65 ± 0.18 | 71.02 ± 0.16 |
| NUCLEOTIDE TRANSFORMER (HUMAN) | 55.65 ± 0.18 | 75.59 ± 0.15 |
| NUCLEOTIDE TRANSFORMER (MULTI-SPECIES) | 53.71 ± 0.18 | 77.76 ± 0.14 |
| EVO2 | – | **78.98 ± 0.15** |

*Table A8.* Binary classification of functional ROIs versus strand-aware proximal upstream regions (balanced accuracy, %). Values are mean ± SE.

| Model | Random Init (%) | Pretrained (%) |
|---|---|---|
| ARGAMBA NTP-ONLY | 59.39 ± 0.17 | 66.99 ± 0.17 |
| ARGAMBA CEP-ONLY | 59.84 ± 0.17 | 70.85 ± 0.16 |
| ARGAMBA NTP+CEP | 59.85 ± 0.17 | 71.40 ± 0.16 |
| BI-GAMBA MLM-ONLY | 59.78 ± 0.17 | 65.92 ± 0.17 |
| BI-GAMBA MEM-ONLY | 59.54 ± 0.17 | **72.90 ± 0.16** |
| BI-GAMBA MLM+MEM | 59.63 ± 0.17 | 72.65 ± 0.16 |
| CADUCEUS | 62.51 ± 0.17 | 70.03 ± 0.16 |
| HYENADNA | 54.15 ± 0.17 | 36.52 ± 0.14 |
| PHYLOGPN | 54.94 ± 0.18 | 66.40 ± 0.17 |
| NUCLEOTIDE TRANSFORMER (HUMAN) | 45.04 ± 0.18 | 69.57 ± 0.16 |
| NUCLEOTIDE TRANSFORMER (MULTI-SPECIES) | 46.44 ± 0.18 | 72.06 ± 0.16 |
| EVO2 | – | **74.62 ± 0.16** |
| K-mer model ($k = 6$) | – | 62.75 ± 0.17 |
| PhyloP (6D) | – | 69.55 ± 0.15 |

*Table A9.* Variant effect prediction on ClinVar pathogenic vs. benign missense variants (AUROC). Top two values overall are bolded.

| Model | Log-likelihood | Pred. cons. |
|---|---|---|
| ARGAMBA NTP-ONLY | 0.528 | – |
| ARGAMBA CEP-ONLY | – | 0.585 |
| ARGAMBA NTP+CEP | 0.538 | 0.595 |
| BI-GAMBA MLM-ONLY | 0.476 | – |
| BI-GAMBA MEM-ONLY | – | 0.641 |
| BI-GAMBA MLM+MEM | 0.495 | 0.635 |
| **PhyloP** | – | **0.804** |
| **Evo2** | **0.856** | – |

*Table A10.* Variant effect prediction on COSMIC frequent-in-cancer vs. common missense variants (AUPRC). Top two values overall are bolded.

| Model | Log-likelihood | Pred. cons. |
|---|---|---|
| ARGAMBA NTP-ONLY | 0.028 | – |
| ARGAMBA CEP-ONLY | – | 0.014 |
| ARGAMBA NTP+CEP | 0.013 | 0.017 |
| BI-GAMBA MLM-ONLY | 0.022 | – |
| BI-GAMBA MEM-ONLY | – | 0.022 |
| BI-GAMBA MLM+MEM | 0.013 | 0.023 |
| **PhyloP** | – | **0.086** |
| **Evo2** | **0.064** | – |

*Table A11.* Variant effect prediction on GWAS fine-mapped causal vs. matched missense variants (AUPRC). Top two values overall are bolded.

| Model | Log-likelihood | Pred. cons. |
|---|---|---|
| ARGAMBA NTP-ONLY | 0.114 | – |
| ARGAMBA CEP-ONLY | – | 0.172 |
| ARGAMBA NTP+CEP | 0.119 | 0.175 |
| BI-GAMBA MLM-ONLY | 0.126 | – |
| BI-GAMBA MEM-ONLY | – | **0.190** |
| BI-GAMBA MLM+MEM | 0.118 | 0.174 |
| **PhyloP** | – | **0.215** |
| **Evo2** | 0.168 | – |

*Table A12.* Variant effect prediction on OMIM non-coding pathogenic vs. common variants (AUPRC). Top two values overall are bolded.

| Model | Log-likelihood | Pred. cons. |
|---|---|---|
| ARGAMBA NTP-ONLY | 0.089 | – |
| ARGAMBA CEP-ONLY | – | 0.158 |
| ARGAMBA NTP+CEP | 0.091 | 0.156 |
| BI-GAMBA MLM-ONLY | 0.093 | – |
| BI-GAMBA MEM-ONLY | – | 0.155 |
| BI-GAMBA MLM+MEM | 0.095 | 0.157 |
| **PhyloP** | – | **0.673** |
| **Evo2** | **0.381** | – |

*Table A13.* Variant effect prediction on gnomAD rare pathogenic vs. common missense variants (AUROC). Top two values overall are bolded.

| Model | Log-likelihood | Pred. cons. |
|---|---|---|
| ARGAMBA NTP-ONLY | 0.454 | – |
| ARGAMBA CEP-ONLY | – | 0.589 |
| ARGAMBA NTP+CEP | 0.452 | 0.587 |
| BI-GAMBA MLM-ONLY | 0.428 | – |
| BI-GAMBA MEM-ONLY | – | 0.591 |
| BI-GAMBA MLM+MEM | 0.426 | **0.593** |
| **PhyloP** | – | **0.611** |
| **Evo2** | – | – |

*Table A14.* Variant effect prediction on GWAS fine-mapped non-coding causal vs. matched variants (AUPRC). Top two values overall are bolded.

| Model | Log-likelihood | Pred. cons. |
|---|---|---|
| ARGAMBA NTP-ONLY | 0.099 | – |
| ARGAMBA CEP-ONLY | – | 0.138 |
| ARGAMBA NTP+CEP | 0.100 | 0.139 |
| BI-GAMBA MLM-ONLY | 0.101 | – |
| BI-GAMBA MEM-ONLY | – | 0.143 |
| BI-GAMBA MLM+MEM | 0.097 | **0.146** |
| **PhyloP** | – | **0.212** |
| **Evo2** | 0.106 | – |

*Table A15.* Held-out (test split) vs. training performance across all evaluation tasks (balanced accuracy, %). Reported values are mean $\pm$ standard error. $\Delta$ BA = test $-$ training.

| Model | Split | Random (%) | Upstream (%) | Rnd-noannot (%) | MC-ROI (%) | MC-ROI100bp (%) |
|---|---|---|---|---|---|---|
| ARGAMBA NTP-ONLY | train | $71.34 \pm 0.18$ | $66.76 \pm 0.19$ | $71.51 \pm 0.18$ | $53.88 \pm 0.21$ | $36.24 \pm 0.22$ |
| | test | $70.64 \pm 0.35$ | $64.40 \pm 0.37$ | $70.42 \pm 0.35$ | $52.69 \pm 0.38$ | $36.72 \pm 0.47$ |
| | $\Delta$ | $-0.70$ | $-2.36$ | $-1.09$ | $-1.19$ | $+0.48$ |
| ARGAMBA CEP-ONLY | train | $75.54 \pm 0.17$ | $70.81 \pm 0.18$ | $75.51 \pm 0.17$ | $49.98 \pm 0.23$ | $37.76 \pm 0.23$ |
| | test | $74.12 \pm 0.34$ | $70.56 \pm 0.35$ | $74.46 \pm 0.34$ | $48.98 \pm 0.43$ | $37.22 \pm 0.44$ |
| | $\Delta$ | $-1.42$ | $-0.25$ | $-1.05$ | $-1.00$ | $-0.54$ |
| ARGAMBA NTP+CEP | train | $76.18 \pm 0.17$ | $71.67 \pm 0.18$ | $76.79 \pm 0.17$ | $53.09 \pm 0.23$ | $39.74 \pm 0.23$ |
| | test | $75.26 \pm 0.34$ | $70.24 \pm 0.36$ | $75.60 \pm 0.34$ | $52.18 \pm 0.43$ | $38.62 \pm 0.43$ |
| | $\Delta$ | $-0.92$ | $-1.43$ | $-1.19$ | $-0.91$ | $-1.12$ |
| BI-GAMBA MLM-ONLY | train | $71.12 \pm 0.18$ | $66.05 \pm 0.19$ | $71.62 \pm 0.18$ | $34.65 \pm 0.22$ | $23.44 \pm 0.21$ |
| | test | $69.80 \pm 0.35$ | $64.67 \pm 0.37$ | $70.82 \pm 0.35$ | $32.72 \pm 0.40$ | $22.66 \pm 0.41$ |
| | $\Delta$ | $-1.32$ | $-1.38$ | $-0.80$ | $-1.93$ | $-0.78$ |
| BI-GAMBA MEM-ONLY | train | $77.08 \pm 0.17$ | $72.96 \pm 0.18$ | $77.33 \pm 0.17$ | $41.92 \pm 0.24$ | $33.39 \pm 0.22$ |
| | test | $75.78 \pm 0.33$ | $71.78 \pm 0.35$ | $76.63 \pm 0.32$ | $40.87 \pm 0.47$ | $32.98 \pm 0.45$ |
| | $\Delta$ | $-1.30$ | $-1.18$ | $-0.70$ | $-1.05$ | $-0.41$ |
| BI-GAMBA MLM+MEM | train | $76.75 \pm 0.17$ | $72.61 \pm 0.18$ | $77.17 \pm 0.16$ | $41.92 \pm 0.23$ | $33.31 \pm 0.22$ |
| | test | $75.58 \pm 0.33$ | $71.84 \pm 0.35$ | $77.11 \pm 0.31$ | $39.53 \pm 0.43$ | $31.93 \pm 0.42$ |
| | $\Delta$ | $-1.17$ | $-0.77$ | $-0.06$ | $-2.39$ | $-1.38$ |

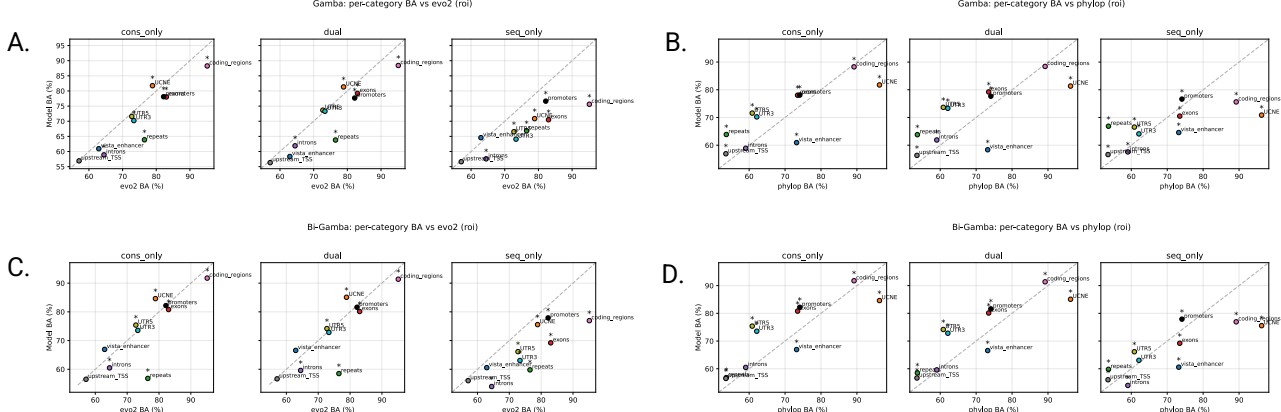

*Figure A5.* Per-category balanced accuracy (BA) comparison between learned representations and PhyloP or Evo2 on the upstream localization task. Each point corresponds to a functional category, with the x-axis showing either Evo2 (panels A,C) or PhyloP (panels B,D) BA and the y-axis showing Gamba model BA, evaluated on strand-aware functional ROIs versus proximal upstream regions. The dashed line indicates parity. Points above the line indicate categories where the learned representation outperforms the reference. Stars denote categories where the BA difference is significant after Bonferroni correction across categories.

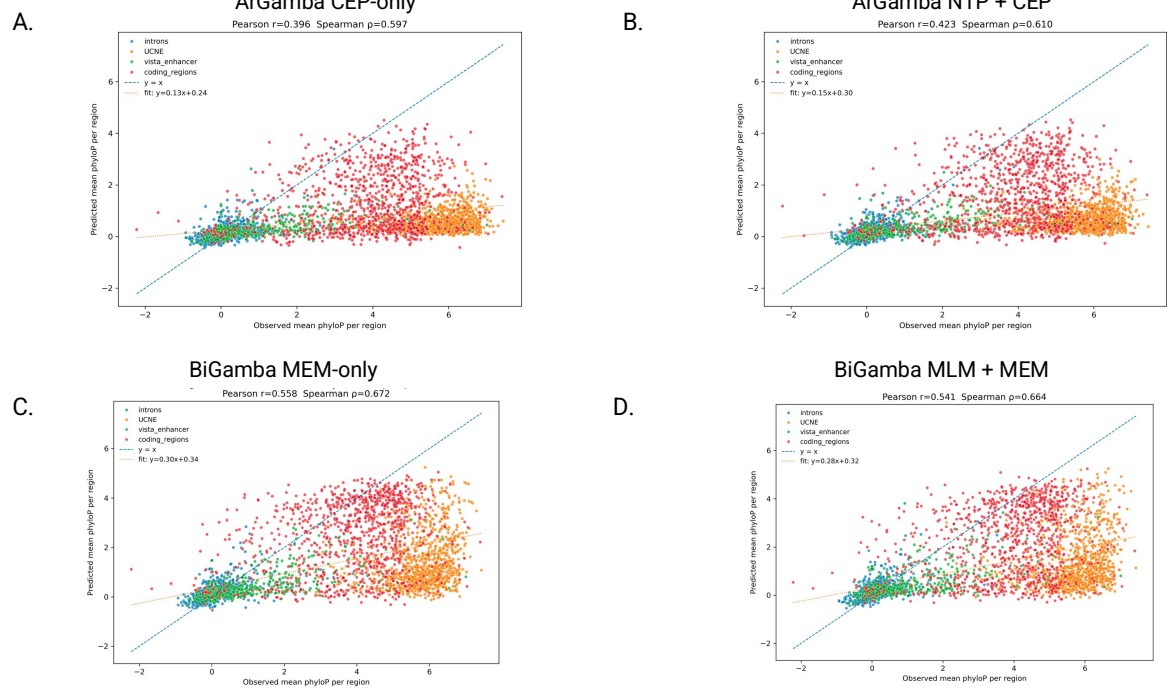

*Figure A6.* Region-level correlation with different genomic regions true and predicted PhyloP scores.

*Table A16.* Classification of ATG codon contexts. Values are mean $\pm$ SE from leave-one-out 1-nearest neighbor classification.

| Model | Random Init (%) | Pretrained (%) |
|---|---|---|
| ARGAMBA NTP-ONLY | $36.67 \pm 5.88$ | $44.59 \pm 5.82$ |
| ARGAMBA CEP-ONLY | $37.11 \pm 6.09$ | $51.87 \pm 6.56$ |
| ARGAMBA NTP+CEP | $36.81 \pm 6.19$ | $53.50 \pm 8.63$ |
| BI-GAMBA MLM-ONLY | $39.08 \pm 6.73$ | $46.94 \pm 7.12$ |
| BI-GAMBA MEM-ONLY | $39.59 \pm 6.78$ | $56.22 \pm 11.97$ |
| BI-GAMBA MLM+MEM | $38.69 \pm 6.63$ | $56.16 \pm 11.42$ |
| CADUCEUS | $41.99 \pm 6.43$ | $59.46 \pm 7.34$ |
| PHYLOGPN | $42.83 \pm 6.75$ | $74.24 \pm 5.92$ |
| NUCLEOTIDE TRANSFORMER (HUMAN) | $22.50 \pm 5.89$ | $55.87 \pm 4.85$ |
| NUCLEOTIDE TRANSFORMER (MULTI-SPECIES) | $20.16 \pm 5.42$ | $\mathbf{80.11 \pm 3.62}$ |
| EVO2 | – | $74.41 \pm 3.63$ |
| K-mer model ($k = 6$) | – | $33.11 \pm 4.03$ |
| K-mer model ($k = 3$) | – | $27.91 \pm 24.03$ |
| PhyloP (6D) | – | $54.26 \pm 10.64$ |

