# OpenReview forum: "Predicting evolutionary rate as a pretraining task improves genome language model representations"
_ICML.cc/2026/Conference — ICML 2026 regular_

### Official Review · Reviewer_eL7i · 2026-02-22

**Soundness:** 3
**Presentation:** 3
**Significance:** 3
**Originality:** 2
**Overall Recommendation:** 5
**Confidence:** 5

**Summary:**

This paper describes an evaluation of whether including prediction of PhyloP scores (taken from0 a multiple sequence alignment of 241 mammalian genomes) as a pretraining task improves the performance of human genome language models (gLMs) at various genome annotation and interpretation tasks including predicting annotation of functional regions (promoters, enhancers, ultra-conserved elements, repeats, and various features of protein-coding genes from GENCODE); predicting start codons; and predicting variant effects. Benchmarks are constructed, sorted on statistics like region length and GC content, and several models are evaluated including Caduceus, Nucleotide Transformer, and a few others. Pretraining is evaluated on sequence only, PhyloP scores only, or both sequence and PhyloP. The benchmarks demonstrate that inclusion PhyloP scores improves performance at these tasks.

**Compliance With Llm Reviewing Policy:**

Affirmed.

**Final Justification:**

Rebuttal nudged score upwards.

**Key Questions For Authors:**

Have you considered discussing in more detail how PhyloP scores are constructed, and what kind of signal those scores might retain, omit, or distort?

Is there a reason you didn’t evaluate GPN-STAR and PhyloGPN?

Why not try some non-human language models?

**Limitations:**

yes

**Strengths And Weaknesses:**

This is a competent translation of a natural idea (that inclusion of evolutionary information, summarized via point substitution rates estimated from a multiple alignment, can inform language models of biological sequences) into a benchmark organized for machine learning researchers. As such it will probably be useful, and used. The selection of evaluation tasks is reasonably comprehensive, although much of it seems to be taken from the datasets collated by the Song lab. The positive results are not very surprising, but it is good to have a concrete data-founded demonstration that this approach has promise.

The biggest weaknesses center on the fact that this is a fairly superficial exploration of the potential of using evolutionary information. There is a lot more that one might hope for from this kind of approach. Everything here smacks of “take this idea and quickly produce something that will be useful to the ML community” with very little in the way of an effort to explore the best way to do this. The authors acknowledge this themselves in the Conclusion, stating:

“While we use PhyloP scores derived from
241 mammalian species from the Zoonomia project (Con-
sortium, 2020), future work could expand to alternative
scores, including PhastCons scores (Zhou et al., 2011),
which capture evolutionary rates over broader contexts, and
PhyloP scores from alternative MSAs, such as vertebrate
alignments, which have been shown to outperform the 241-
mammalian–derived PhyloP scores on variant effect predic-
tion (Benegas et al, 2024).”

Indeed so. It would be good to explore this point in considerably more detail. For example, despite the paper’s frequent conflation of PhyloP scores with “evolutionary rates”, these scores actually represent a highly processed statistic derived from evolutionary rates: specifically, PhyloP scores are log-odds ratios derived on a maximum likelihood estimate of evolutionary rate, with a sign adjustment based on whether the estimated rate is higher or lower than a neutral baseline. (The authors do mention the sign adjustment.)

PhastCons scores are closer to actual substitution rates, but it is not quite correct to say that they “capture evolutionary rates over broader contexts”. All of these scores are estimated from multiple sequence alignments (MSAs) and generally the deeper (and more reliable) the alignment, the more information will be captured. This is not really acknowledged beyond a citation to (Benegas et al, 2024) to the point that vertebrate alignments contribute more than the mammalian subset to variant effect prediction.

More interestingly, the approach of fitting continuous-time finite-state Markov chain models of nucleotide substitution on phylogenetic factor graphs to individual columns of an MSA, while trusty and reliable, probably misses a lot. The authors do cite GPN-MSA, a transformer-based approach which steps away from the single-column constraint to attend to a region of the MSA around the focal column of interest. It would have been interesting to see how this compares to the fine-tuned single-sequence models developed here. PhyloGPN (another related approach) is included in the benchmark, which is useful. GPN-STAR is not.

Given GPN-MSA, GPN-STAR, and PhyloGPN (all Song lab approaches which directly incorporate phylogenies and MSAs), it is a bit hard to endorse some of the apparent claims to priority (e.g. in the Abstract “These results establish evolution as a key training target for genome-scale models”, or in the Introduction “despite its effectiveness in detecting function, evolutionary rate has not yet been widely investigated for training gLMs.”) To be fair, these claims are carefully worded, and the other approaches are cited - though not all of them are included in the benchmark and it’s not totally clear why.

---

> ### Author Rebuttal · Authors · 2026-03-30
>
> We thank the reviewer for their comments and address their concerns below.
>
> **Superficial exploration of evolutionary information**
>
> The simplicity of our approach was deliberate, as our goal was to show even a simple, suboptimal use of evolutionary scores produces substantial effects. We share the reviewer’s concerns with the tendency to take an idea from bioinformatics and run with it quickly, rather than thoroughly investigate it. We agree that it is not surprising that evolutionary information improves gLMs, but we believe the magnitude of effect is surprising, given our simple set-up. Our small models are nearly competitive with SOTA models two to three orders of magnitude larger on our functional region of interest tasks.
> As we noted to the other reviewers, we did extensively experiment with alternative loss strategies (reviewer vgf9) and normalization (reviewer RL2p), but excluded these when we found they had little impact. We intend to include many of these previously-completed experiments in the Supplementary of the camera-ready version. We also hope that the design of our evaluations demonstrates our engagement with the underlying biology. As we discussed in response to reviewer vgf9, we re-designed benchmarks to control for confounders not previously considered.
>
>
>
> **Specifically, PhyloP scores are log-odds ratios derived on a maximum likelihood estimate of evolutionary rate**
>
> PhyloP scores are signed -log_10(p-values) from a likelihood ratio test (LRT). A scaling factor is fit at each alignment column using MLE, but the final score is derived from comparing this fitted model to neutral evolution (null model) via LRT, with the sign indicating conservation or acceleration.
>
> **PhastCons scores are closer to actual substitution rates**
>
> Our methodology uses PhyloP scores over PhastCons for two reasons. One, its single-base resolution matches the information density of sequence reconstruction tasks exactly, allowing us to compose our prediction task with the single-sequence pretraining tasks most gLMs are trained with. Two, it captures conservation and acceleration signals, whereas PhastCons are biased toward detecting conserved elements. We agree both scores discard information relative to the full MSA and will discuss this limitation, including alignment depth constraints, in the camera-ready version.
>
>
> **Apparent claims to priority are hard to endorse**
>
> We will improve the clarity of these statements by amending them to say “These results establish the relative effectiveness of evolution as a key training target for genome-scale models compared to sequence-based methods” and “despite its effectiveness in detecting function, isolating the effect of training on evolutionary rate has not yet been investigated for training gLMs”.
>
>
> 1. **Discuss how PhyloP scores are constructed and what signal they retain, omit, or distort**
>
> We agree this warrants more discussion. In the camera-ready version, we plan to discuss trade-offs between scores in more detail, as well as signals likely missed across all scores that require sequence alignments. PhyloP scores don’t handle indels, the rate heterogeneity over the evolutionary tree,  or asymmetry in the rate of evolutionary depending on direction of substitution. In addition, relative to MSAs, it is lossy of information like the distribution over nucleotides at a position.
>
> 2. **Is there a reason you didn’t evaluate GPN-STAR and PhyloGPN?**
>
> We note that we did evaluate PhyloGPN and assume the reviewer meant GPN-MSA. GPN-STAR and GPN-MSA require MSA inputs, unlike single-sequence methods evaluated here. We scope around single-sequence models, which are still critical to assess as firstly, they are composable with our methodology, and secondly, they have different properties than models that rely upon MSAs - for example, they can be applied to synthetic sequences, whereas MSA-dependent models cannot. We will more explicitly note that MSA-based models achieve superior VEP performance in the camera-ready version.
>
> 3. **Why not try some non-human language models?**
>
> We do benchmark several multi-species models, motivated by theory from protein language models, suggesting such models implicitly learn frequent patterns over evolution, in contrast to our explicit use of PhyloP scores as a target. Our focus on the human genome is consistent with the scope of many existing genome language models (e.g. Caduceus), where the aim is to deepen understanding of human/mammalian regulatory genomics. There are many impactful applications of human/mammalian-only genome language models (for example predicting disease variants and designing synthetic enhancers).

---

> > ### Author Rebuttal · Reviewer_eL7i · 2026-03-31
> >
> > Thank you

---

> > > ### Author Response · Authors · 2026-04-01
> > >
> > > Thank you - if you believe our response fully resolves the issues you raised, would you consider raising your score?

---

### Official Review · Reviewer_RL2p · 2026-03-12

**Soundness:** 3
**Presentation:** 3
**Significance:** 2
**Originality:** 3
**Overall Recommendation:** 3
**Confidence:** 4

**Summary:**

This paper proposes a novel pre-training approach for genome language models (gLMs) designed to better capture biological and regulatory signals. The framework integrates standard natural language processing-inspired sequence reconstruction with novel tasks that predict the rate of evolution. A composed training objective allows for a controlled comparison of models trained on sequence only, evolutionary rate only, or both jointly. The framework is evaluated on a newly developed suite of biologically grounded benchmarks and established variant effect prediction tasks, demonstrating that joint pre-training enables relatively small models to outperform sequence-only baselines and compete with much larger existing gLMs.

**Compliance With Llm Reviewing Policy:**

Affirmed.

**Key Questions For Authors:**

1. Can you explicitly confirm whether the zero-shot evaluations (e.g., functional region separation) were strictly limited to the held-out test chromosomes (2 and 22)? If they were sampled from the entire hg38 genome, how do you account for the possibility of data leakage?
2. What percentage of the 1 billion base-pair training corpus lacked a PhyloP score and was consequently set to zero? How did you ensure the model did not trivially learn to predict zero to minimize loss?
3. Did you experiment with normalizing the PhyloP scores? Given the distribution shown in Figure 1, normalization seems like it could improve model stability and performance.
4. Could you provide a clearer justification for using Jamba and MoE for a 66M parameter model? Specifically, how do the state-space components benefit the processing of shorter sequences?
5. Given that pre-training was restricted to hg38, have you evaluated the representations on alternative species (e.g., plants, bacteria) to see if the evolutionary signals learned from human data generalize across distant phylogenetic clades?

**Limitations:**

See my comments

**Strengths And Weaknesses:**

### Strengths

- Embedding evolutionary site-rate estimates directly into the pre-training objective is an interesting and biologically sound direction for genomic language models (gLMs).
- The experimental design is commendable. Pre-training on both next-token prediction and masked-language-modeling allows for a direct comparison between the two primary training paradigms, and embedding several methods for integrating site-wise estimation within each paradigm allows for direct comparisons, which strengthen the evaluation.
- The empirical evaluations are thorough, effectively demonstrating where the inclusion of evolutionary rates is beneficial. The claims are fair and not overstated.

### Weaknesses

- The choice to use a Jamba/Mixture of Experts (MoE) architecture is not well-justified given the scale and domain. MoE is typically leveraged to scale up massive models efficiently, but at a mere 66M parameters, the model could easily be run densely without the MoE overhead. Furthermore, state-space architectures are primarily designed to handle ultra-long contexts; if downstream tasks involve only a 2,000 nucleotide context window, this architectural choice seems unnecessary.
- The model is pre-trained on the human reference genome (hg38) and heavily evaluated on tasks derived from the human genome. Unless the evaluation Regions of Interest (ROIs) are strictly constrained to the held-out test chromosomes, there is a severe risk of data leakage, meaning the models may simply be memorising the pre-training distribution rather than learning generalizable biological signals.
- The authors state that positions without a PhyloP score are set to zero. However, they do not report the ratio of missing scores. If a vast majority of the sequence lacks scores, the model may achieve artificially low loss simply by collapsing to predict zero most of the time. Furthermore, the lack of score normalization (varying from -1 to 7.22) may destabilize training.
- Limiting the pre-training dataset entirely to the human genome inherently biases the model toward mammalian genomics, potentially limiting its utility for diverse biological domains (e.g., bacteria or plants). The downstream analysis only uses mammalian data, and does not embed plant, vertebrate or bacterium datasets. Without this, it is hard to say if the model is biased or not.

---

> ### Author Rebuttal · Authors · 2026-03-30
>
> Thank you.
>
> 1. **Confirm zero-shot evals used held-out chr2/22 only; otherwise address leakage risk**
>
> Below is a table with the delta in ROI classification performance for balanced accuracy between training and test chromosomes. There is negligible difference between the two (1-2%), and the relative ranking of models within each family is identical across training and held-out splits, confirming that the benefits of evolutionary rate prediction are not dependent on the choice of split.
>
> ### Held-out vs Training Performance (Δ = test − training, balanced accuracy %)
> | Model | Random (%) | Upstream (%) | Rnd-noannot (%) | MC-ROI (%) | MC-ROI100bp (%) |
> |---|---|---|---|---|---|
> | **ARGAMBA NTP-ONLY** | −0.70 | −2.36 | −1.09 | −1.19 | +0.48 |
> | **ARGAMBA CEP-ONLY** | −1.42 | −0.25 | −1.05 | −1.00 | −0.54 |
> | **ARGAMBA NTP+CEP** | −0.92 | −1.43 | −1.19 | −0.91 | −1.12 |
> | **BI-GAMBA MLM-ONLY** | −1.32 | −1.38 | −0.80 | −1.93 | −0.78 |
> | **BI-GAMBA MEM-ONLY** | −1.30 | −1.18 | −0.70 | −1.05 | −0.41 |
> | **BI-GAMBA MLM+MEM** | −1.17 | −0.77 | −0.06 | −2.39 | −1.38 |
>
> We reported numbers across all chromosomes in the submission for benchmarking fairness, as other models use different splits, do not report hold-out splits, or do not hold out any of the human genome. We recognize this raises data leakage concerns, so in the camera-ready version, we will include these results as a Supplementary table (anonymized link to table with results across splits [here](https://drive.google.com/file/d/11dI8ujjCVpkNdrXHAM2HWEbQHCKJBT-3/view?usp=sharing)), and will note inconsistent hold-out practices as a field-wide limitation.
>
> 2. **What % of training corpus lacked PhyloP scores (set to zero), and how was trivial zero-prediction prevented?**
>
> A similar issue was also raised by reviewer 1, vgf9. After our filtering of the data prior to training, across the 1,398,602,778 base pairs available for training, testing and validation, only 2,565,127 (0.18%) PhyloP scores were missing. We believe this negligible fraction of regions with missing values are unlikely to skew the model's estimates, and furthermore, reflect the effectiveness of our data-cleaning step.
>
> 3. **Did you experiment with normalizing PhyloP scores?**
>
> We also Gaussian normalized the PhyloP scores, but it did not provide notable improvements in training stability, and even resulted in less-accurate PhyloP score predictions. With Bi-Gamba models at 190k steps on held-out data, the unnormalized model achieves Pearson r=0.727 and Spearman ρ=0.689, while the Gaussian-normalized variant performs slightly worse (Pearson r=0.716, Spearman ρ=0.679), with a shallower regression slope (0.45 vs 0.50), we therefore retained the raw scale. (Anonymized link [here](https://drive.google.com/file/d/1yQsDFYIwwBZMPgtcjbSNla2IoX849pfa/view?usp=sharing) to figure).
>
> Consistent with our response to reviewer vgf9 about weighted versions of our loss, we excluded these results from the submission version of our work, because we wanted to emphasize that our results held even in the most simple possible set-up (and were not the consequence of elaborate weighing or pre-processing strategies). We now realize from these exchanges that it is worth documenting these additional experiments to demonstrate that our methods were established through empirical tests.
>
> 4. **Justify using Jamba/MoE at 66M params; how do state-space components benefit shorter sequences?**
>
> This decision was made to establish models for future scaling, especially as biological sequence models are increasingly employing these architectures (Yang et al., The DayHoff Atlas, 2025). Evo2 establishes that SSMs scale well for genomes, and in future work we want to model longer sequences. Having strong results at lower parameter counts is beneficial to set up future work even if the architecture is currently overpowered for our setup.
>
>
> 5. **Evaluation of representations on non-human species to test generalization across phylogenetic clades**
>
> Our focus on the human genome is consistent with the scope of many existing genomic foundation models (e.g. Caduceus), where the aim is to deepen understanding of human/mammalian regulatory genomics. Within these bounds there are many impactful applications (for example predicting disease variants, designing synthetic enhancers), that show a model is valuable even if it doesn’t generalize outside of mammalian regulatory architectures.
>
> The reviewer’s point advances an interesting question of if our strategy of training on evolutionary conversation is general in that we will see benefits across biological domains, versus just for humans. PhyloP scores have been employed to analyze plant, vertebrate, and bacterial genomes, but different evolutionary dynamics make transfer from predictions on the 240-mammalian alignment unclear. Extending this strategy to novel biological domains should thus be validated in the future, and we will additionally flag this as critical to investigate.

---

> > ### Author Rebuttal · Reviewer_RL2p · 2026-04-06
> >
> > Thank you for your clarification. Nevertheless, it did not remove the two issues that matter for my previous recommendation.
> > - First, I'm still not convinced for leakage-resistant for ATG/VEP evaluation.
> > - Second, the central claim still collapses to a much narrower statement than the paper advertises.

---

> > > ### Author Response · Authors · 2026-04-06
> > >
> > > We thank the reviewer for their comments.
> > >
> > > We would like to clarify for the first point that both evolutionary rate-trained variants (NTP+ CEP, CEP-only and MLM+MEM and MEM-only) outperform sequence-only training on the ATG task for both ArGamba and Bi-Gamba, in both the full and held-out chromosome evaluations. Although the ordering between NTP+ CEP, CEP-only and MLM+MEM, MEM-only can vary, the broader conclusion that adding evolutionary-rate supervision improves over sequence-only training remains unchanged regardless of the split.
> > >
> > > For ATG:
> > >
> > > | Family   | Training task | Full (%) | Held-out (%) | Δ (pp) |
> > > | -------- | ------------- | -------: | -----------: | -----: |
> > > | Bi-Gamba | MLM-only      |    36.11 |        38.30 |  +2.19 |
> > > | Bi-Gamba | MEM-only     |    43.89 |        45.72 |  +1.83 |
> > > | Bi-Gamba | MLM+MEM          |    44.57 |        45.61 |  +1.04 |
> > > | ArGamba    | NTP-only      |    34.90 |        36.99 |  +2.09 |
> > > | ArGamba    | CEP-only     |    43.11 |        41.08 |  -2.03 |
> > > | ArGamba    | NTP+CEP          |    44.81 |        43.64 |  -1.17 |
> > >
> > > For the VEP task, the held-out split preserves the same qualitative conclusion for most tasks: the best evolutionary rate-trained variant remains better than sequence-only for nearly all model-family/task comparisons with available held-out results. The main exception is the COSMIC missense evaluation, which favours sequence-only for ArGamba in both splits and for Bi-Gamba in the held-out split.
> > >
> > > For VEP:
> > > | Family   | Training task | VEP task                                       |  Full  | Held-out |      Δ |
> > > | -------- | ------------- | ---------------------------------------------- | ----: | -------: | -----: |
> > > | ArGamba  | NTP-only      |  ClinVar pathogenic vs benign missense  | 0.528 |    0.527 | -0.001 |
> > > | ArGamba  | CEP-only      | ClinVar pathogenic vs benign missense | 0.585 |    0.568 | -0.017 |
> > > | ArGamba  | NTP+CEP       | ClinVar pathogenic vs benign missense  | 0.595 |    0.577 | -0.019 |
> > > | Bi-Gamba | MLM-only      | ClinVar pathogenic vs benign missense | 0.476 |    0.480 | +0.004 |
> > > | Bi-Gamba | MEM-only      | ClinVar pathogenic vs benign missense    | 0.641 |    0.618 | -0.023 |
> > > | Bi-Gamba | MLM+MEM       | ClinVar pathogenic vs benign missense    | 0.635 |    0.612 | -0.023 |
> > > | ArGamba  | NTP-only      | COSMIC frequent in cancer vs common missense | 0.028 |    0.147 | +0.119 |
> > > | ArGamba  | CEP-only      | COSMIC frequent in cancer vs common missense | 0.014 |    0.020 | +0.006 |
> > > | ArGamba  | NTP+CEP       | COSMIC frequent in cancer vs common missense | 0.017 |    0.046 | +0.029 |
> > > | Bi-Gamba | MLM-only      | COSMIC frequent in cancer vs common missense | 0.022 |    0.089 | +0.067 |
> > > | Bi-Gamba | MEM-only      | COSMIC frequent in cancer vs common missense | 0.022 |    0.025 | +0.003 |
> > > | Bi-Gamba | MLM+MEM       | COSMIC frequent in cancer vs common missense | 0.023 |    0.035 | +0.012 |
> > > | ArGamba  | NTP-only      | GWAS finemapped missense causal vs matched   | 0.114 |    0.093 | -0.021 |
> > > | ArGamba  | CEP-only      | GWAS finemapped missense causal vs matched   | 0.172 |    0.140 | -0.032 |
> > > | ArGamba  | NTP+CEP       | GWAS finemapped missense causal vs matched   | 0.175 |    0.153 | -0.022 |
> > > | Bi-Gamba | MLM-only      | GWAS finemapped missense causal vs matched   | 0.126 |    0.143 | +0.017 |
> > > | Bi-Gamba | MEM-only      | GWAS finemapped missense causal vs matched   | 0.190 |    0.167 | -0.023 |
> > > | Bi-Gamba | MLM+MEM       | GWAS finemapped missense causal vs matched   | 0.174 |    0.150 | -0.024 |
> > > | ArGamba  | NTP-only      | OMIM noncoding pathogenic vs common | 0.089 |    0.097 | +0.009 |
> > > | ArGamba  | CEP-only      | OMIM noncoding pathogenic vs common | 0.158 |    0.217 | +0.059 |
> > > | ArGamba  | NTP+CEP       | OMIM noncoding pathogenic vs common  | 0.156 |    0.223 | +0.066 |
> > > | Bi-Gamba | MLM-only      | OMIM noncoding pathogenic vs common | 0.093 |    0.095 | +0.003 |
> > > | Bi-Gamba | MEM-only      | OMIM noncoding pathogenic vs common | 0.155 |    0.167 | +0.012 |
> > > | Bi-Gamba | MLM+MEM       | OMIM noncoding pathogenic vs common  | 0.157 |    0.188 | +0.030 |
> > > | ArGamba  | NTP-only      | GWAS finemapped | 0.099 |    0.106 | +0.006 |
> > > | ArGamba  | CEP-only      | GWAS finemapped  | 0.138 |    0.144 | +0.006 |
> > > | ArGamba  | NTP+CEP       | GWAS finemapped   | 0.139 |    0.139 | +0.001 |
> > > | Bi-Gamba | MLM-only      | GWAS finemapped | 0.101 |    0.104 | +0.003 |
> > > | Bi-Gamba | MEM-only      | GWAS finemapped  | 0.143 |    0.150 | +0.008 |
> > > | Bi-Gamba | MLM+MEM       | GWAS finemapped  | 0.146 |    0.151 | +0.005 |
> > >
> > > We believe this addresses the central concern of data-leakage since our claims hold across splits for both tasks.
> > >
> > > Furthermore, with respect to the reviewer's second point, we can amend a camera-ready version to be more explicit that our central claims pertain to the human genome.

---

### Official Review · Reviewer_vgf9 · 2026-03-13

**Soundness:** 3
**Presentation:** 3
**Significance:** 3
**Originality:** 3
**Overall Recommendation:** 4
**Confidence:** 3

**Summary:**

The paper introduces the Gamba model family, to improve gLM representations by incorporating evolutionary rate prediction directly into the pretraining process. The authors propose two tasks - Current Evolution Prediction (CEP) and Masked Evolution Modeling (MEM) , where models learn to predict PhyloP scores (a measure of evolutionary conservation) alongside standard sequence reconstruction. Utilizing hybrid Transformer-Mamba and bidirectional Mamba architectures, the approach is evaluated on a custom suite of zero-shot benchmarks testing the models' ability to distinguish functional genomic regions and recognize the genetic code without task-specific training. Results show that models trained with evolutionary signals outperform sequence-only baselines and achieve competitive performance with models orders of magnitude larger, such as the 7B Evo2, particularly in identifying functional regions and predicting the effects of genetic variants.

**Compliance With Llm Reviewing Policy:**

Affirmed.

**Final Justification:**

The rebuttal was helpful in clarifying the intended scope and strengthened the paper’s significance and clarity, particularly around the biological motivation and the relevance of the zero-shot and VEP settings. I continue to view the work as interesting, but my main soundness concern remains, the current evidence supports the narrower claims in the paper more convincingly than the broader claim of general pretraining superiority. Overall, the rebuttal reinforced rather than changed my prior assessment, so I am keeping my original score.

**Key Questions For Authors:**

- How did the model handle positions where PhyloP scores were unavailable? Did the zero-initialization strategy force the model to predict neutrality for unaligned regions, and did you mask these positions during loss calculation?
- Why do the models struggle significantly more with Enhancers and Promoters than with Exons in the upstream-localization task? Is this a limitation of the single-nucleotide resolution of the training signal?
- In the absence of a weighting hyperparameter for the combined loss, how did you ensure that the magnitude of the Gaussian NLL loss did not overwhelm the sequence reconstruction gradients during early training?

**Limitations:**

No major concerns on the ethical or societal implications of the work.

**Strengths And Weaknesses:**

**Strengths:**
- The paper is clearly written and is easy to follow. The authors have also shared the code and implementation details
- The paper addresses the critical problem of the junk DNA signal-to-noise ratio in genomics by move beyond simple sequence reconstruction to learn the functional grammar of the genom. The use of PhyloP scores as a dense, single-base teacher is a sound approach for anchoring a language model in biological constraints.
- The results demonstrate that incorporating biological priors allows relatively small models (4M to 66M parameters) to achieve competitive performance with massive state-of-the-art models like Evo2.
- The authors design a new benchmark with harder tasks, such as distinguishing a functional region from its own 2kb upstream sequence
- The results show that the models is able to predict evolutionary rates on held-out test chromosomes, indicating they have learned general biological patterns. Additionally, the ablation studies are well set up.


**Weaknesses:**

- Genomic foundation models typically benchmark across many tasks. There is no evaluation on BEND, GUE, NT benchmark, DART-Eval. Without standard benchmarks, it is difficult to verify the claims outside of this specific experimental setup.
- While the model learns to predict evolutionary rates, the predicted scores are consistently weaker predictors than the ground-truth PhyloP scores it was trained on. This might suggest that the model is a lossy approximation of its training labels rather than a deeper biological interpreter. If a model is significantly smaller or less trained than current SOTA models, it is difficult to distinguish whether the performance gaps are due to the pretraining objective itself or simply a lack of computational capacity
- Results from the ATG-separation task indicate that evolutionary pretraining did not really help the model learn the genetic code or triplet reading frame logic as effectively as multi-species sequence models. Here, Evo2 performs best, while Gamba models are competitive but not dominant.
- Between ArGamba and BiGamba, because there are differences in the architecture, training objective and model sizes, it is not clear how we can isolate the effect of directionality or model size cleanly?
- Loss is a simple sum with no weighting or scheduling and is not explored. While acknowledged by the authors, this is an important aspect especially when discussing about pretraining strategies

---

> ### Author Rebuttal · Authors · 2026-03-30
>
> Thank you.
>
> **Without standard benchmarks, hard to verify claims**
>
> All benchmarks listed except Dart-Eval (which has had limited adoption) have recently been shown to be better solved by random weights, questioning their utility for assessing if gLMs are learning biological signals (Vishniakov et al., 2025). We designed benchmarks robust to these issues: zero-shot, directly testing pre-training impact without fine-tuning confounds, and controlling for biological confounders absent from the listed benchmarks. Here randomly initialized models do not achieve strong performance relative to pretrained gLMs, confirming sufficient task difficulty. We did employ the VEP benchmark (Ye et al.,2025) for our evaluations, which is more challenging, and like our benchmarks, operates zero-shot.
>
> **Models are weaker predictors than PhyloP scores they’re trained on**
>
> While this holds for the VEP task, Gamba models consistently and significantly outperform PhyloP on the functional region tasks (see Figure 2, Supplementary Figure A1). This suggests that the value of predicting evolution rate is not simply approximating these labels. Gamba’s relative VEP task weakness could be a lack of computational capacity; current models could be learning coarse region-level signal and not fine-grained nucleotide-level effects. Scaling experiments are reserved for future work; our focus is demonstrating that evolutionary rate pretraining tasks are efficient and composable with sequence-based ones.
>
> **Method underperforms multi-species models on ATG-separation**
>
> We agree that evolutionary rate pretraining provided only modest gains in learning the genetic code. Our work demonstrates trade-offs between pretraining strategies, so we do not think it is essential that our evolutionary-rate pretrained models are SOTA, or that evolutionary-rate pretraining is superior to sequence reconstruction in all settings.
>
> To our knowledge, we are the first to systematically benchmark to what extent gLMs learn the genetic code. These results are surprising and suggest that evolutionary-rate reconstruction may be learning an alternative route to understanding exons compared to sequence-based pretraining, and also suggests why composing the two pretraining tasks is effective.
>
> **Can’t isolate the effect of directionality or model size**
>
> We agree; our goal was to isolate evolutionary rate pretraining, not compare directionality or model size. Bidirectional/unidirectional variants were included to confirm results held across common gLM pretraining strategies, and further disentanglement is beyond our current scope.
>
> **Loss is a simple sum**
>
> We experimented with a focal Gaussian Negative Likelihood Loss (GNLL) (adapted from Lin et al.,2017), which upweights nucleotides with high residuals to address the rarity of high magnitude PhyloP scores. Both losses produced nearly identical sequence reconstruction losses (see anonymized link [here](https://drive.google.com/file/d/1MoeDHGMBSeV_rZ966S1pPm4HN97WVb5Y/view?usp=sharing)), and did not consistently improve predictions (anonymized link [here](https://drive.google.com/file/d/1EVZW4-OXHldkzzoCI-zW7OUhFkxCTvsN/view?usp=sharing) to MSE calculated from predicted means). In the interest of demonstrating that our results held true in the most naive case, we report the simpler regular GNLL. We will include experiments on the focal GNLL in the Supplementary of the camera-ready.
>
> 1. **Clarify missing PhyloP scores, zero-init bias, and loss masking**
>
> Missing scores were set to 0 as noted in Section 3.3. They were not masked. Across the 1,398,602,778 bp available for training, testing and validation, only 2,565,127 (0.18%) PhyloP scores were missing. This reflects the effectiveness of our data cleaning, and we will update the camera-ready version to include these details.
>
> 2. **Why upstream localization underperforms on enhancers/promoters vs exons?**
>
> We do attribute this to limited single-nucleotide resolution (as evidenced by our VEP task). This is also implied by Gamba’s strong performance on UCNEs, which have high conservation for the entire region, whereas conservation within our VISTA enhancers occurs within small windows. In the camera-ready, we will use this observation to further expand on a discussion of how existing gLMs (including Gamba) have limited learning of nucleotide-level signal relative to region-level signal.
>
> 3. **Did Gaussian NLL dominate reconstruction loss in early training?**
>
> Loss curves for the sequence reconstruction (CE loss) for ArGamba NTP+CEP and ArGamba NTP-only are similar across early training (anonymized link [here](https://drive.google.com/file/d/1_XVE45Y4DAJH4PlwEJCGpUztJHYPf8BN/view?usp=sharing)). ArGamba NTP+CEP has a converged CE loss of 1.303, compared to 1.289 for NTP-only. For Bi-Gamba, MLM+MEM has a CE loss of 1.180 and MLM-only of 1.176. The losses are not significantly different from other gLMs on our validation dataset: Caduceus achieves a MLM CE loss of 1.178.

---

> > ### Author Rebuttal · Reviewer_vgf9 · 2026-04-03
> >
> > Thank you for the rebuttal. The motivation is compelling, and I agree that the proposed evaluation suite is stronger and more biologically grounded than many standard gLM benchmarks. That said, while the evaluation is strong, I do not think it is fully complete enough to support the paper’s claims. In particular, I believe the results support the narrower claim that evolutionary-rate pretraining is a useful and biologically meaningful signal for the proposed zero-shot and VEP settings, but I do not think the current evaluation fully establishes broader superiority as a general gLM pretraining strategy. Additionally, some questions remain open regarding how much of the observed behavior is attributable to the pretraining objective itself versus limitations in model scale or training capacity. Overall, I think this is an interesting paper, and I would like to maintain a positive view of it. I would therefore like to keep my original score.

---

### Decision · Program_Chairs · 2026-04-30

**Decision:**

Accept (regular)

**Comment:**

This paper studies whether predicting evolutionary rate can serve as a useful pretraining task for genome language models, and evaluates this idea through the proposed Gamba family on zero-shot functional region benchmarks and variant effect prediction. The reviews agreed that the core idea is biologically meaningful and technically interesting. Reviewers also found the evaluation suite more grounded than many standard genomic benchmarks, and the strongest positive feedback centered on the finding that relatively small models trained with evolutionary supervision can outperform sequence-only baselines and remain competitive with much larger models in several settings.

The main concerns were about the scope of the paper's claims rather than the basic validity of the approach. In particular, one reviewer remained unconvinced that the evidence fully rules out leakage concerns in all headline comparisons. Reviewers also argued that the paper more strongly supports a narrower claim, namely that evolutionary-rate pretraining is useful in the proposed zero-shot and VEP settings, than a broader claim of general pretraining superiority. Other reviewers raised related questions about how much of the observed behavior should be attributed to the objective itself rather than model scale or other design choices.

The rebuttal addressed many of these points. The authors clarified the handling of missing PhyloP scores, provided additional evidence addressing leakage concerns, explained the architectural choices more clearly, and added supporting experiments on loss design and normalization. These responses were sufficient for one reviewer to mark their concerns as fully resolved and for another to raise their score. At the same time, the weak-reject reviewer was not fully persuaded, where they continued to view the leakage-resistant evaluation as incomplete and the paper's strongest claim as broader than the evidence currently supports.

While I lean to think the technical merits outweigh the remaining concerns, I agree that the strongest version of the paper's claim should be stated more carefully in the final version. The final version of this submission is required to sharpen the scope of its claims, make the leakage-related evaluation discussion as clear as possible, and more explicitly distinguish the supported empirical message from broader claims of general pretraining superiority.